# Charge-driven condensation of RNA and proteins suggests broad role of phase separation in cytoplasmic environments

Bercem Dutagaci[1], Grzegorz Nawrocki[1], Joyce Goodluck[2], Ali Akbar Ashkarran[3], Charles G Hoogstraten[1], Lisa J Lapidus[2]*, Michael Feig[1]*

[1]Department of Biochemistry and Molecular Biology, Michigan State University, East Lansing, United States; [2]Department of Physics, Michigan State University, East Lansing, United States; [3]Precision Health Program and Department of Radiology, Michigan State University, East Lansing, United States

**Abstract** Phase separation processes are increasingly being recognized as important organizing mechanisms of biological macromolecules in cellular environments. Well-established drivers of phase separation are multi-valency and intrinsic disorder. Here, we show that globular macromolecules may condense simply based on electrostatic complementarity. More specifically, phase separation of mixtures between RNA and positively charged proteins is described from a combination of multiscale computer simulations with microscopy and spectroscopy experiments. Phase diagrams were mapped out as a function of molecular concentrations in experiment and as a function of molecular size and temperature via simulations. The resulting condensates were found to retain at least some degree of internal dynamics varying as a function of the molecular composition. The results suggest a more general principle for phase separation that is based primarily on electrostatic complementarity without invoking polymer properties as in most previous studies. Simulation results furthermore suggest that such phase separation may occur widely in heterogenous cellular environment between nucleic acid and protein components.

*For correspondence:
lapidus@msu.edu (LJL);
mfeiglab@gmail.com (MF)

**Competing interests:** The authors declare that no competing interests exist.

## Introduction

Biological cells compartmentalize to support specific functions such as stress response (*Boulon et al., 2010*; *Protter and Parker, 2016*), regulation of gene expression (*Boisvert et al., 2007*; *Morimoto and Boerkoel, 2013*), and signal transduction (*Su et al., 2016*). Compartmentalization by organelles that are surrounded by lipid membranes is well known. In addition, membrane-less organelles that result from coacervation have been described (*Alberti et al., 2019*; *Banani et al., 2017*; *Boeynaems et al., 2018*; *Ditlev et al., 2018*). In the nucleus, they include the nucleolus (*Feric et al., 2016*; *Iarovaia et al., 2019*), nuclear speckles (*Galganski et al., 2017*; *Lamond and Spector, 2003*), and cajal bodies (*Cioce and Lamond, 2005*; *Gall, 2000*; *Machyna et al., 2013*); stress granules (*Protter and Parker, 2016*; *Burke et al., 2015*; *Molliex et al., 2015*); germ granules (*Brangwynne et al., 2009*; *Voronina et al., 2011*); and processing bodies *Fromm et al., 2014*; *Luo et al., 2018* have been found in the cytoplasm. The formation of coacervates via condensation and phase separation depends on the composition and concentration of the involved macromolecules (*Ditlev et al., 2018*) as well as environmental conditions such as pH, temperature, and the concentration of ions (*Alberti et al., 2019*; *Elbaum-Garfinkle et al., 2015*; *Ruff et al., 2018*). Multivalent interactions, the presence of conformationally flexible molecules (*Sawyer et al., 2019*; *Radhakrishna et al., 2017*; *de Kruif et al., 2004*), and electrostatic interactions between highly charged molecules (*Sawyer et al., 2019*; *de Kruif et al., 2004*; *Fay and Anderson, 2018*; *Cummings and Obermeyer, 2018*; *Michaeli et al., 1957*;

*Mattison et al., 1995*) are well-known as the key factors that promote phase separation (PS), in particular via complex coacervation (*Sing, 2017*; *Andreev et al., 2018*). In biological environments, nucleic acids such as RNA have been found to play a prominent role in condensate formation due to their charge (*Chujo et al., 2016*; *Clemson et al., 2009*; *Falahati et al., 2016*; *Mitrea et al., 2016*; *Smith et al., 2016*; *Van Treeck et al., 2018*; *Garcia-Jove Navarro et al., 2019*). Another component often found in biological condensates are intrinsically disordered peptides (IDPs) that may phase separate alone or in combination with RNA (*Ruff et al., 2018*; *Smith et al., 2016*; *Brady et al., 2017*; *Dignon et al., 2018a*; *Posey et al., 2018*), although disorder may not be essential for phase separation (*Sanders et al., 2020*; *Aumiller et al., 2016*). Condensates often materialize as droplets, where experiments such as fluorescence recovery after photobleaching (FRAP) (*Shin et al., 2017*; *Taylor et al., 2019*) or direct visualization of merging droplets (*Van Treeck et al., 2018*; *Li et al., 2012*) may confirm liquid-like behavior. However, a variety of other types of less-liquid condensates involving biomolecules have been described including clusters, gels, and aggregation to fibrils or tangles (*Molliex et al., 2015*; *Alberti and Hyman, 2016*; *Weber, 2017*; *Weber and Brangwynne, 2012*; *Lin et al., 2015*; *Jain et al., 2016*). In those cases, internal diffusional dynamics may be highly retarded or lost. The high degree of polydispersity in biological multicomponent systems presents additional changes. An especially intriguing aspect of polydisperse systems is the propensity for multiphasic behavior (*Feric et al., 2016*; *Sanders et al., 2020*; *Lu and Spruijt, 2020*), which imparts a potential for fine-grained tunable spatial patterning of biomolecules in cellular systems (*Sanders et al., 2020*).

Biomolecular condensates have been studied extensively (*Mitrea et al., 2018a*). Microscopy (*Elbaum-Garfinkle et al., 2015*; *Banani et al., 2016*; *Nott et al., 2015*), nuclear magnetic resonance (NMR) spectroscopy (*Burke et al., 2015*; *Brady et al., 2017*), fluorescence spectroscopy (*Feric et al., 2016*; *Mitrea et al., 2018b*; *Wei et al., 2017*), X-ray diffraction (*Kato et al., 2012*; *Lin et al., 2016a*), and scattering methods *Mitrea et al., 2018b*; *Li et al., 2012*; *Riback et al., 2017* have characterized in vitro (*Feric et al., 2016*; *Burke et al., 2015*; *Elbaum-Garfinkle et al., 2015*; *Brady et al., 2017*; *Nott et al., 2015*; *Wei et al., 2017*; *Kato et al., 2012*; *Lin et al., 2016a*; *Riback et al., 2017*) and in vivo systems (*Brangwynne et al., 2009*; *Brangwynne et al., 2011*; *Maharana et al., 2018*). Theoretical studies have complemented experiments (*Mitrea et al., 2018a*; *Dignon et al., 2019*), including particle-based simulations (*Dignon et al., 2018b*) and analytical approaches based on polymer (*Posey et al., 2018*; *Brangwynne et al., 2015*) and colloid theories (*Nguemaha and Zhou, 2018*; *Qin and Zhou, 2017*; *Woldeyes et al., 2017*). Additional insights into specific interactions have come from molecular dynamics (MD) simulation studies (*Wei et al., 2017*; *Rauscher and Pomès, 2017*; *Pak et al., 2016*). Polymer aspects of IDPs and unstructured RNA were emphasized in applications of Flory-Huggins theory in combination with simulations (*Feric et al., 2016*; *Dignon et al., 2018b*; *Fei et al., 2017*; *Lin et al., 2016b*). Related studies in the colloid field have described the phase behavior of macromolecules and nanoparticles as single spherical particles (*Nguemaha and Zhou, 2018*; *Qin and Zhou, 2017*; *Woldeyes et al., 2017*). However, most of the latter studies so far have focused on liquid-solid transitions and the formation of finite size clusters in monodisperse systems. Despite progress, it has remained unclear what components can lead to condensation, especially in highly heterogeneous cellular environments.

As most previous studies have focused on specific biomolecules undergoing PS, we focus here on the question of how general of a phenomenon PS may be in biological environments and what factors may determine the propensity for PS in a heterogeneous system. The starting point is a molecular model of a bacterial cytoplasm that was established by us previously (*Feig et al., 2015*; *Yu et al., 2016*) and that was simulated here again but using colloid-like spherical particles with a potential parameterized against atomistic MD simulations of concentrated protein solutions. Coarse-grained modeling of cytoplasmic environments has a long history of impressive earlier efforts (*Ridgway et al., 2008*; *McGuffee and Elcock, 2010*; *Ando and Skolnick, 2010*; *Wang and Cheung, 2012*; *Xu et al., 2013*; *Hasnain et al., 2014*; *Trovato and Tozzini, 2014*; *Bicout and Field, 1996*) as reviewed in more detail elsewhere (*Feig et al., 2017*; *Feig and Sugita, 2019*). However, the time and spatial scales covered here are more extensive than in previous work, allowing us to focus on PS processes. We found that distinct phases enriched with highly negatively charged RNA and positively charged proteins were formed in the simulations, consistent with a generic electrostatic mechanism that does not require specific interaction sites or elements of disorder and may apply broadly to mixtures of nucleic acids and proteins. The phase behavior seen in the cytoplasmic system was

reproduced in reduced five- and two-component models and described by an analytical model where we could systematically vary molecular charge, size, and concentrations. The main prediction of the formation of condensates between RNA and positively charged proteins was confirmed experimentally via confocal microscopy and FRET spectroscopy and the nature of the condensates was analyzed further via dynamic light scattering and nuclear magnetic resonance spectroscopy. The details of the findings from simulation, theory, and experiment are described in the following.

## Results

### Condensates enriched in tRNA and ribosomes form in a model bacterial cytoplasm

A model of the cytoplasm of *Mycoplasma genitalium* established previously (*Feig et al., 2015*; *Yu et al., 2016*) was simulated at a coarse-grained (CG) level with one sphere per macromolecule or complex (*Supplementary file 1*). CG particle interactions were calibrated against results from atomistic MD simulations of concentrated protein solutions. The parameters involve only two particle-dependent properties, namely size and charge. Droplet-like condensates formed spontaneously within 20 μs (*Figure 1A/B*) and remained present during 1 ms simulation time. Similar results were obtained with an alternate effective charge model that resulted in better agreement between theory and experiment (see below; *Figure 1—figure supplement 1*) Two types of condensates were observed: one type contained predominantly tRNAs and positively charged proteins; the other type contained ribosome particles (RP) and positively charged proteins. The RP condensates also attracted the weakly negatively charged GroEL particles at the surface (*Figure 1A*). The condensates increased in size as the system size was increased from 100 to 300 nm (*Figure 1A*). This observation is consistent with PS rather than finite-size cluster formation. The presence of multiple droplets in the 300 nm system suggests incomplete convergence, but as the droplets grow in size, further merging becomes kinetically limited due to slowing diffusion. We did not find evidence for growth via Ostwald ripening where particles preferentially evaporate from smaller condensates and redeposit onto larger condensates. Further analysis focused on the condensates observed in the 100 nm system.

Cluster analysis considered interactions between the nucleic acids and positively charged proteins to obtain trajectory-averaged cluster size distributions (*Figure 1C*). Most tRNA (87%) was part of a condensate. The remaining fraction of tRNA existed as monomers or small clusters, suggesting coexistence of dilute and condensed phases. RP were only found in the RP condensates. Total macromolecular volume fractions inside tRNA and RP condensates were 0.42 and 0.58, respectively, whereas volume fractions for just tRNA and RP inside their respective condensates were 0.07 and 0.26. The volume of the condensates was estimated based on the overlapping van der Waals volumes of spheres inside the largest cluster with an additional probe of 2.2 nm in consistent with our cluster definition. The dilute phase volume was estimated as the remaining accessible volume after subtracting volume of condensates from the total volume. The condensates had significantly higher macromolecular densities than the rest of the simulated system (*Figure 1—figure supplement 2*). The moderately high volume fractions for tRNA condensates are still within the range of concentrated liquid phases (*Dumetz et al., 2008*), but the higher volume fractions in the RP condensate tend toward solid- or gel-like phases (*Dumetz et al., 2008*). Radial distribution functions of tRNA and RP from the center of the corresponding condensates show a relatively smooth decay with a soft boundary for tRNA condensates (*Figure 1—figure supplement 3*), that are consistent with a more dynamic phase, whereas distinct peaks and a sharper boundary for RP indicate a highly ordered arrangement in the RP condensates. The more ordered structure of the RP condensates may be an example of the kind of structured condensates resulting from a balance between homotypic and heterotypic interactions as described recently (*Regy et al., 2020*).

We observed separate condensates involving tRNA or RP, presumably due to the large difference in size of RP vs. tRNA that may be explained at least in part by the Asakura-Oosawa depletion model (*Asakura and Oosawa, 1958*). Both tRNA and RP condensates contained (positively charged) proteins at high concentrations. tRNA and RP interactions with those proteins were favorable as evidenced by a strong peak in the pairwise radial distribution function $g(r)$ at contact distance (*Figure 1—figure supplement 4*). The charge and size of the proteins attracted to the condensates

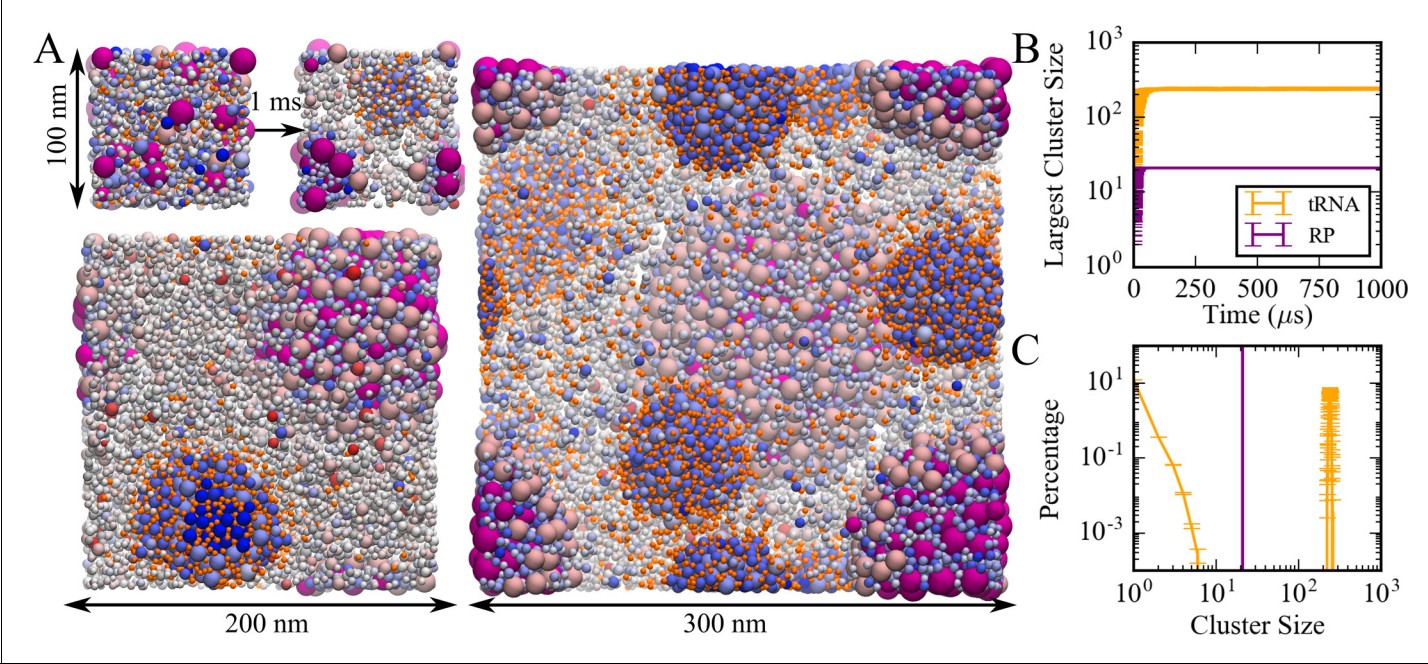

**Figure 1.** Coarse-grained simulations of a model bacterial cytoplasm. (**A**) Initial and final frames for 100 nm box and final frames for 200 and 300 nm boxes are shown with tRNAs in orange, ribosomes in magenta, and other molecules colored according to their charges (blue toward positive charges; red toward negative charges). Sphere sizes are shown proportional to molecular sizes. Large pink spheres correspond to GroEL particles. (**B**) Size of the largest cluster vs. simulation time in 100 nm system. (**C**) Cluster size distributions for tRNA and RP during the last 500 μs in the 100 nm system. The online version of this article includes the following figure supplement(s) for figure 1:

**Figure supplement 1.** Coarse-grained simulations of a model bacterial cytoplasm with an alternative effective charge model using Equation 6.

**Figure supplement 2.** Density variation in the cytoplasmic model system during the last 500 μs of the simulation.

**Figure supplement 3.** Radial distribution curves for tRNA and RP in condensates from the center of their respective condensates.

**Figure supplement 4.** Pairwise radial distribution functions between tRNA, RP, and positively charged protein particles and any other particles in the cytoplasmic model system.

**Figure supplement 5.** Number of proteins in the tRNA and RP condensates vs. the radius (**A**) and charge (**B**) of the proteins found in the condensates.

**Figure supplement 6.** Mean square displacement (MSD) for tRNA (left) and RP (right) during the first and last 1 μs of the cytoplasmic simulations.

**Figure supplement 7.** Translational diffusion of macromolecules in the cytoplasmic system as a function of the radius of the macromolecules during the first and last 1 μs of the simulations.

**Figure supplement 8.** Comparison of effective charge models that take into counterion condensation according to *Equation 5* (orange) or *Equation 6* (blue) for moderate (**A**) and high (**B**) nominal charges.

**Figure supplement 9.** An illustrative example of packing of a tRNA pair (red) in close contact with the positively charged proteins (pink) and other tRNAs (blue) in the cytoplasmic simulations based on the last snapshot after 1 ms simulation.

**Figure supplement 10.** Radial distribution functions for tRNA-tRNA interactions in the five-component model system with different POS$_L$ radii in comparison with the cytoplasmic system.

**Figure supplement 11.** Radial distribution functions of protein-protein pairs (top) and cluster size distributions for proteins (bottom) in simulations of mixtures of villin, protein G, and ubiquitin at volume fractions of 5, 10, and 30%.

**Figure supplement 12.** The tRNA cluster at the final snapshot of the cytoplasmic system.

**Figure supplement 13.** Histograms of tRNA cluster sizes for the cytoplasmic system using the geometrical clustering and based on pairwise contacts using different distance cutoffs added to σ$_{ij}$.

differed between tRNA and RP condensates (*Figure 1—figure supplement 5*). In the tRNA condensates, large proteins with radii of 3 nm and above and with charges of 10 and above were preferred. In contrast, the proteins in the RP condensates were smaller, with radii of 3 nm or less, and many proteins had charges below 10. This suggests that differential interactions between different size and charge nucleic acid and protein particles may further explain the formation of separate condensates involving tRNA and RP.

The dynamics inside and outside the condensates was analyzed in terms of translational diffusion coefficients ($D_{tr}$) calculated based on mean-squared displacements (*Figure 1—figure supplement*

6). Diffusion during the last 1 μs of the simulation was compared with diffusion during the first 1 μs when condensates were not yet formed. Molecule-specific values of $D_{tr}$ are given in *Supplementary file 1*. As a function of the radius of the macromolecules (*Figure 1—figure supplement 7*), $D_{tr}$ values follow a similar trend as observed before in atomistic simulations of the same system. Diffusion outside the condensates resembled diffusion in the dispersed phase. In tRNA condensates, the diffusion of macromolecules is similar to the dispersed phase or is moderately retarded, depending on the molecule, and consistent with reduced diffusion in increased protein concentrations seen in experiment (*Muramatsu and Minton, 1988*; *Zimmerman and Minton, 1993*). In RP condensates, diffusion is reduced to a greater extent, but significant dynamics is still maintained for most types of macromolecules as they diffuse around a relatively static RP cluster (*Video 1*).

## Factors promoting RNA condensation in a reduced five-component model system

A simplified five-component system was constructed to reproduce the RNA condensation observed in the cytoplasmic model. The simplified model consisted of tRNA, ribosome particles (RP), large (POS$_L$, $q = 20$, $r = 3.5$ nm) and small (POS$_S$, $q = 1$, $r = 2.52$ nm) positively charged proteins as well as neutral crowders (CRW, $q = 0$, $r = 2.52$ nm). tRNA and RP concentrations were initially set as in the cytoplasmic model while concentrations, sizes, and charges of the other three particle types were adjusted to match the total number of particles, total molecular volume, and total charge of the cytoplasmic system as closely as possible. Subsequently, a series of simulations were run at different concentrations and with different parameters (*Supplementary file 2*).

In simulations of the five-component model, tRNA and RP condensed separately as in the cytoplasmic model (*Figure 2—figure supplement 1*). Again, the condensates formed quickly, within 50 μs (*Figure 2—figure supplement 1*), and cluster size distributions of tRNA and RP resembled the results from the cytoplasmic system (*cf. Figure 1* and *Figure 2—figure supplement 1*). However, in contrast to the cytoplasmic system, we found a small fraction (2% on average) of RP in the dilute phase. As in the cytoplasmic model, tRNA strongly preferred interactions with the larger POS$_L$ particles, whereas RP interacted favorably with both POS$_S$ and POS$_L$ (*Figure 2—figure supplement 2*). tRNA condensates remained highly dynamic as in the cytoplasmic system. From the last 100 μs of the simulation, we obtained diffusion coefficients $D_{tr}$ for tRNA of 28.3 ± 0.7 and 59.0 ± 0.5 nm²/μs inside and outside of the condensates, respectively, similar to values of 16.3 ± 0.1 and 55.5 ± 0.8 nm²/μs in the cytoplasmic system. Diffusion coefficients for RP inside and outside of the RP condensates were 0.49 ± 0.01 and 0.80 ± 0.4 nm²/μs, respectively, compared to $D_{tr}$ = 0.34 ± 0.01 nm²/μs for RP in the cytoplasmic condensates.

RP and POS$_L$ concentrations were varied systematically, while the concentration of POS$_S$ was kept constant and the number of CRW particles was adjusted to maintain a constant total molecular volume (*Supplementary file 2*). Cluster size distributions were extracted (*Figure 2—figure supplement 3*) and the fraction of tRNA and RP in the large clusters was determined (*Figure 2*). Some degree of clustering occurs at all concentrations, but condensation requires that a significant fraction of particles is found in the largest clusters. Based on a criterion that at least half of the particles are found in one or few large clusters, tRNA and RP condensation occurs for [POS$_L$]>100 μM (*Figure 2*).

Increasing [RP] reduces the amount of tRNA in the tRNA condensates and effectively raises the critical POS$_L$ concentration above which tRNA forms condensates (*Figure 2*). This can be

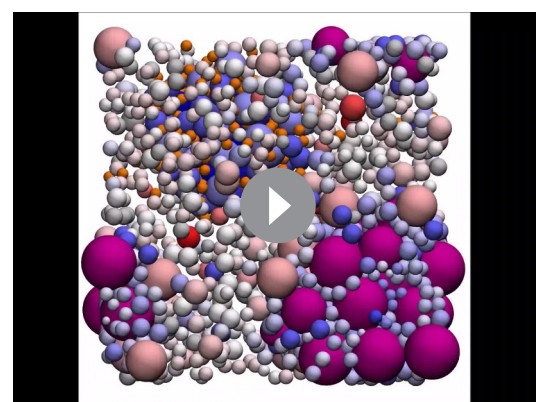

**Video 1.** Simulation of bacterial cytoplasm model. Trajectory of the 100 nm system during the last 1 μs of a 1 ms simulation with tRNAs in orange, ribosomes in magenta, and other molecules colored according to their charges (blue toward positive charges; red toward negative charges). Sphere sizes are shown proportional to molecular sizes. Large pink spheres correspond to GroEL particles.

https://elifesciences.org/articles/64004#video1

understood from competition for POS$_L$. tRNA only interacts significantly with POS$_L$ (*Figure 2—figure supplement 4*) and needs POS$_L$ to form condensates, whereas RP interacts with both POS$_S$ and POS$_L$ (*Figure 2—figure supplement 5*) and therefore draws POS$_L$ from tRNA condensates (*Figure 2—figure supplement 6*). For [POS$_L$]>500 µM, the fraction of tRNA particles in the tRNA condensates is relatively constant (*Figure 2*). However, the number of POS$_L$ particles in the condensates increases as the total [POS$_L$] increases (*Figure 2—figure supplement 6*). This results in larger clusters and lower effective [tRNA] in the condensates at the highest values of [POS$_L$] (*Figure 2—figure supplement 7*). The effect of increasing [RP] is again a depletion of POS$_L$ in the tRNA condensates, so that [tRNA] in the condensates increases with [RP] for a given value of [POS$_L$] (*Figure 2—figure supplement 7*).

In the simulations described so far, the total volume fraction of the system was kept constant by reducing the crowder (CRW) concentration as [POS$_L$] and [RP] increased. Therefore, the decrease in [tRNA] inside the condensates with increasing [POS$_L$] could be due to reduced crowder interactions in the condensate environment. To test this further, we reduced [CRW] without changing [POS$_L$]. Reduced [CRW] also led to reduced [tRNA] in the condensate, but the effect is much smaller than when [CRW] is reduced along with an increase in [POS$_L$] (*Figure 2—figure supplement 8*).

In order to construct phase diagrams, simulations of the five-component model phases were carried out at a range of temperatures for selected values of [RP] and [POS$_L$]. Cluster size distributions were extracted (*Figure 3—figure supplements 1–3*) and the volume fractions of tRNA in dilute and condensed phases as a function of temperature were determined based on the number of tRNA outside and inside the largest tRNA clusters. The volume of the condensed phase containing the largest tRNA cluster was calculated as described above. The resulting curves (*Figure 3*) show the typical features of phase diagrams with phase coexistence below critical temperatures $T_c$ of 400–535 K. In the absence of ribosomes, that is, [RP]=0, an increase in [POS$_L$] lowers $T_c$ and narrows the two-phase regime (*Figure 3C*). This is consistent with reentrant phase behavior expected for complex coacervation of a binary mixture. However, in the presence of ribosomes, that is, [RP]=55 µM, $T_c$ increased at the same time as the two-phase regime narrowed with increasing [POS$_L$] (*Figure 3D*). Moreover, when [POS$_L$]=180 µM, near the minimum needed for PS, an increase in [RP] slightly decreased $T_c$ (*Figure 3A/E*), whereas at a higher concentration, that is, [POS$_L$]=880 µM, $T_c$ increased with increasing [RP] up to a maximum at 55 µM before decreasing (*Figure 3B/E*). These observations reflect competition between ribosomes and tRNA for interactions with POS$_L$ and more generally highlight the effects of a complex interplay between interactions in non-binary mixtures that are more representative of biological environments than simple binary mixtures.

## Phase separation in experiments for binary mixtures of globular RNA and proteins

The results presented so far have focused on multi-component systems that were modeled to reflect the density and distribution of particle sizes and charges in cytoplasmic environments. A key prediction is that PS due to complex coacervation may occur for a wide range of nucleic acids and positively charged proteins simply based on electrostatic complementarity. To test this idea experimentally, we now turn to binary mixtures of globular RNA and positively charged proteins. We focused on the 47-nucleotide J345 Varkud satellite ribozyme RNA, that folds into an approximately globular shape (*Bonneau and Legault, 2014*) and that was mixed at high concentration with common proteins with positive charges and varying sizes for which we may expect PS: myoglobin ($q$ = +2), trypsin ($q$ = +6), lysozyme ($q$ = +8), lactate dehydrogenase (LDH; $q$ = +4), and alcohol dehydrogenase (ADH; $q$ = +8). Bovine serum albumin (BSA; $q$ = −17, $r$ = 2.58 nm) was added as a control, for which condensate formation is not expected due to its negative charge.

Imaging via confocal microscopy of dye-labeled RNA (*Figure 4* and *Figure 4—figure supplements 1–6*) shows well-defined fluorescent clusters for mixtures of RNA with trypsin, ADH, lysozyme, and LDH, but not for RNA with myoglobin or BSA. The background fluorescence varies significantly with protein. It is especially high for the mixtures with LDH, suggesting that only a fraction of RNA is participating in the condensates and a larger fraction of RNA remained in the dilute phase.

Individual condensates are relatively small, and many appear to have sizes near or below the diffraction limit of the microscope. For RNA-trypsin mixtures, we clearly observe single droplet-shaped condensates of varying sizes that follow roughly an exponential distribution (*Figure 4—figure supplement 7*). We note that the concentration of Cy3-labeled RNA is only 8 µM, corresponding to 1 in

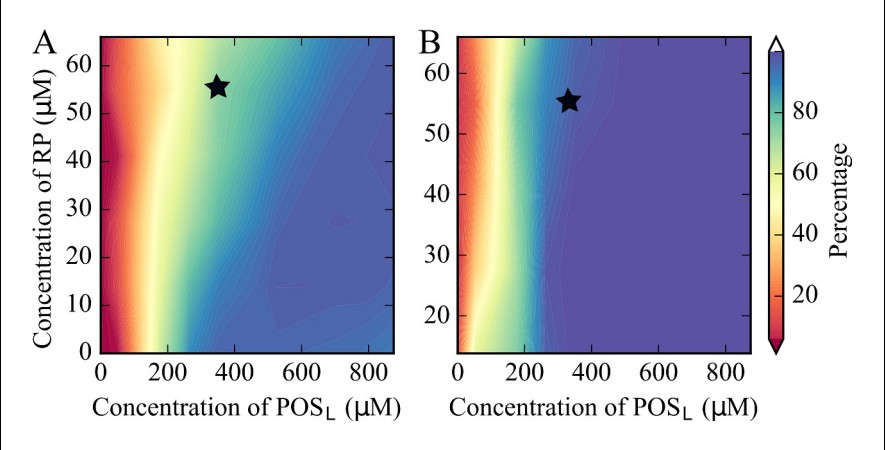

**Figure 2.** Percentage of tRNA (**A**) and RP (**B**) in largest clusters in coarse-grained simulations of the five-component model system as a function of [RP] and [POS$_L$]. The black star indicates the conditions that match the cytoplasmic model. tRNA condensation is a phase separation process.
The online version of this article includes the following figure supplement(s) for figure 2:

**Figure supplement 1.** Initial and final frames of the five-component model system simulation (**A**); time evolution of cluster formation for tRNA and RP clusters (**B**); and cluster size distributions (**C**).
**Figure supplement 2.** Radial distribution functions for interactions between different particle types in the five-component model.
**Figure supplement 3.** Cluster size distribution of tRNA and RP as a function of [RP] and [POS$_L$] in the five-component model system.
**Figure supplement 4.** Radial distribution functions for tRNA with tRNA, POS$_S$, and POS$_L$ as a function of [RP] and [POS$_L$] in the five-component model system.
**Figure supplement 5.** Radial distribution functions for RP with RP, POS$_S$, and POS$_L$ as a function of [RP] and [POS$_L$] concentration in the five-component model system.
**Figure supplement 6.** Relative abundance of POS$_L$ and POS$_S$ in the largest tRNA and RP clusters with the five-component model as a function of [RP] and [POS$_L$].
**Figure supplement 7.** Volume-equivalent radii for largest cluster in tRNA condensates with five-component model (**A**); macromolecular concentrations inside tRNA condensates for tRNA (**B**), POS$_S$ (**C**), and POS$_L$ (**D**).
**Figure supplement 8.** Concentration of tRNA inside the tRNA condensates as a function of [CRW] at constant and increasing values of [POS$_L$] from simulations of the five-component model.
**Figure supplement 9.** Radial distribution functions between tRNA and POS$_S$/POS$_L$ particles in simulations of five-component model at different POS$_L$ concentrations and [RP]=55 µM.
**Figure supplement 10.** Normalized radial distribution functions for tRNA-tRNA (**A**), POS$_L$-POS$_L$ (**B**), tRNA-POS$_L$ (**C**), and POS$_L$-tRNA (**D**) interactions in the condensed (red), dilute (blue), and disperse (green) phases used as input for the theory model.
**Figure supplement 11.** Probability of minimum RNA-RNA distances in the condensed phase from coarse-grained simulations of the five-component model.

---

56 RNA at 0.45 mM total RNA concentration. Therefore, the fluorescent images in *Figure 4* are biased toward clusters that contain at least 50 RNA molecules, whereas smaller clusters are imaged incompletely. RNA-LDH condensates appear similar but we did not attempt a quantitative size analysis due to the high background fluorescence of the RNA-LDH sample. Diffusing droplets in the RNA-trypsin mixture merge over the course of 1 min when they come into proximity (*Figure 4M* and *Videos 2* and *3*), indicative of liquid behavior inside the condensates.

For other proteins (lysozyme and ADH), we found more complex condensate morphologies (*Figure 4*), where smaller condensates associate to form larger, irregular-shaped condensates without merging as seen for RNA-trypsin condensates. This suggests that the condensates with these proteins are less liquid-like, although the exact nature of the condensates not involving trypsin is unclear.

To further study the particle size distributions, we carried out dynamic light scattering (DLS) analysis on RNA/lysozyme and RNA/trypsin samples (*Figure 5*, *Figure 5—figure supplements 1–2*, and

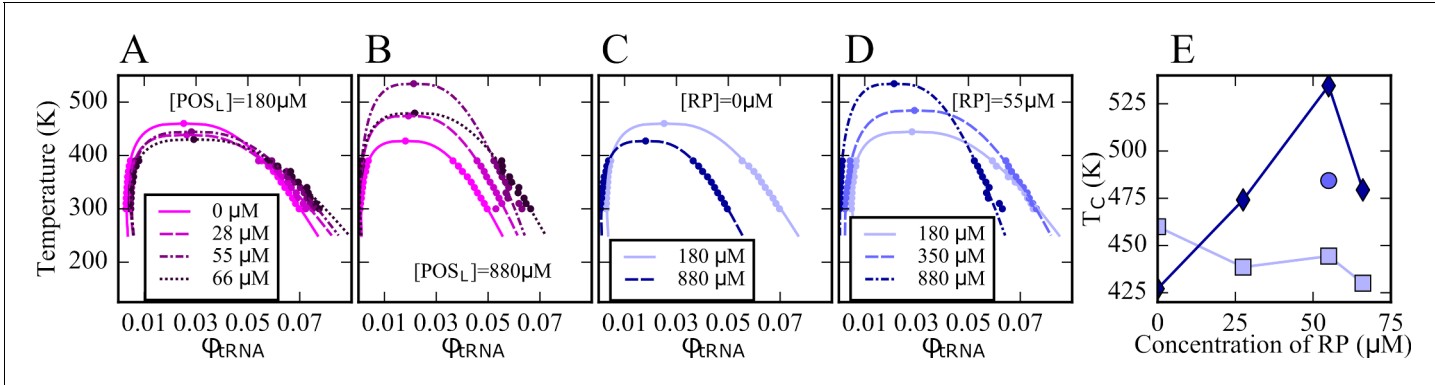

**Figure 3.** Phase diagrams for tRNA with [POS_L]=180 μM and varying RP concentrations (**A**); with [POS_L]=880 μM and varying RP concentrations (**B**); with [RP]=0 at two [POS_L] concentrations (**C**); and with [RP]=55 μM and varying POS_L concentrations (**D**); critical temperatures as a function of [RP] at [POS_L] =180 μM (squares), at [POS_L]=880 μM (diamonds), and at [POS_L]=350 μM (sphere) (**E**). The volume fractions of tRNA in the dilute and condensed phases were obtained based on the number of tRNA particles in the dilute and condensed phases normalized by the respective volumes of the two phases (see Text). Lines in A–D were fitted according to *Equations 9 and 10*.

The online version of this article includes the following figure supplement(s) for figure 3:

**Figure supplement 1.** Cluster size distributions of tRNA at [RP]=55 μM and three POS_L concentrations (see Legend) for temperatures between 300 and 500 K from simulations of the five-component model.

**Figure supplement 2.** Cluster size distributions of tRNA at [POS_L]=180 μM and a range of RP concentrations (see Legend) for temperatures between 300 and 500 K from simulations of the five-component model.

**Figure supplement 3.** Cluster size distributions of tRNA at [POS_L]=880 μM and a range of RP concentrations (see Legend) for temperatures between 300 and 500 K from simulations of the five-component model.

*Table 1*). The light scattering correlation functions indicate a polydisperse sample that is dominated by very long correlation times up to 1 s (*Figure 5*). Those long correlation times theoretically correspond to macroscopic-size particles (*Stetefeld et al., 2016*), but since no such particles were readily visible in the sample, we may conclude that a significant fraction of condensates exhibited very slow diffusion due to surface adsorption. From the correlation function at shorter times, multi-exponential fits suggest particles in two size regimes for RNA-trypsin and in three regimes for RNA-lysozyme. In both cases, the data indicate the presence of 10 nm-scale particles that are consistent with oligomer-size clusters of RNA and protein molecules. Such small clusters between RNA and/or proteins are expected to be present in the dilute phase due to transient associations (*Nawrocki et al., 2017*; *Yildirim et al., 2018*; *Barhoum and Yethiraj, 2010*; *Kowalczyk et al., 2011*). In both, RNA-trypsin and RNA-lysozyme sample, the DLS analysis suggests the presence of μm-size particles (somewhat smaller for trypsin than for lysozyme). In addition, the DLS data indicate the presence of particles at the light microscopy diffraction limit, around 300 nm, for the RNA-lysozyme system but not for RNA-trypsin mixtures. In fact, the DLS results are qualitatively consistent with the microscopy images and provide additional insights into the particle size distributions at and below the light diffraction limit. However, an exact quantitative interpretation of the DLS results is challenging due to the polydispersity and dynamic nature of our samples and for that reason we also did not attempt to quantify what fraction of particles would be expected in the different size regimes.

To map out a phase diagram, we prepared RNA-trypsin mixtures at various experimentally feasible RNA and protein concentrations. PS required a minimum protein concentration, for example with [RNA]=100 μM, PS was found with [trypsin]=150 μM but not with [trypsin]=50 μM (*Figure 6— figure supplement 1*). At the same time, PS was lost when RNA concentrations were too high. The resulting phase diagram based on confocal microscopy imaging is shown in *Figure 6* in comparison with results from theory that are discussed below.

Förster resonance energy transfer (FRET) experiments also showed a significant increase in FRET efficiencies from 50 to 150 μM (*Figure 7A*). The comparison between the microscopy and FRET results furthermore establishes that RNA condensates at this RNA concentration can be recognized by FRET efficiencies above 0.26, whereas lower values may indicate a disperse phase. The gradual increase in FRET efficiencies from 0.24 to 0.26 upon increase of trypsin concentrations from 0 to 50

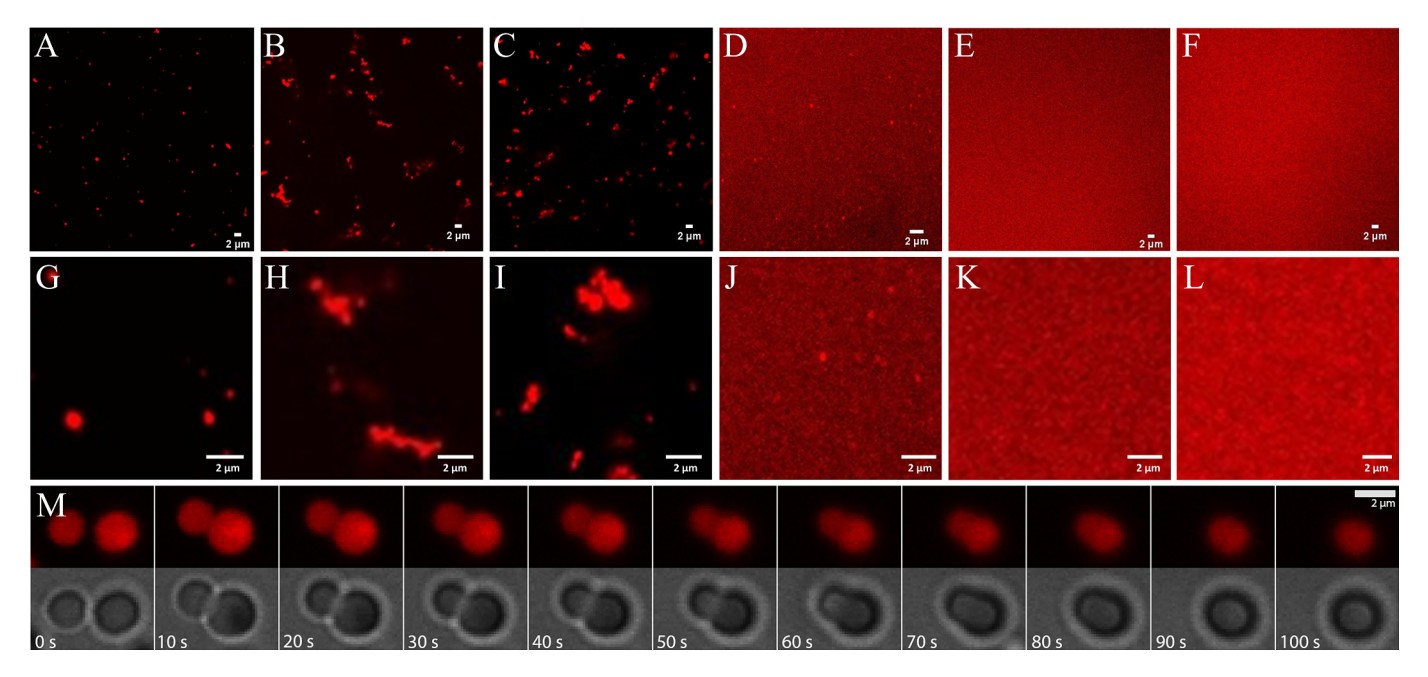

**Figure 4.** Phase separation in mixtures of J345 RNA at 0.45 mM and various globular proteins at 0.35 mM from confocal microscopy of labeled RNA: trypsin (**A**; **G**), ADH (**B**; **H**), lysozyme (**C**; **I**), LDH (**D**; **J**), myoglobin (**E**; **K**), BSA (**F**; **L**). Time lapse of droplet merging in RNA-trypsin mixture from fluorescence and bright-field microscopy imaging (**M**).

The online version of this article includes the following figure supplement(s) for figure 4:

**Figure supplement 1.** Confocal microscopy of labeled J345 RNA for a mixture between J345 RNA at 0.45 mM and trypsin at 0.35 mM.

**Figure supplement 2.** Confocal microscopy of labeled J345 RNA for a mixture between J345 RNA at 0.45 mM and alcohol dehydrogenase at 0.35 mM.

**Figure supplement 3.** Confocal microscopy of labeled J345 RNA for a mixture between J345 RNA at 0.45 mM and lysozyme at 0.35 mM.

**Figure supplement 4.** Confocal microscopy of labeled J345 RNA for a mixture between J345 RNA at 0.45 mM and lactate dehydrogenase at 0.35 mM.

**Figure supplement 5.** Confocal microscopy of labeled J345 RNA for a mixture between J345 RNA at 0.45 mM and myoglobin at 0.35 mM.

**Figure supplement 6.** Confocal microscopy of labeled J345 RNA for a mixture between J345 RNA at 0.45 mM and bovine serum albumin at 0.35 mM.

**Figure supplement 7.** Distribution of cluster sizes from confocal microscopy of labeled J345 RNA in mixtures between J345 RNA at 0.1 mM and trypsin at 0.25 mM.

**Figure supplement 8.** Circular dichroism spectra of trypsin at 0.150 mM (black), J345 RNA, at 0.037 mM (red), and a mixture of trypsin at 0.150 mM and J345 RNA at 0.029 mM (green).

**Figure supplement 9.** 600 MHz $^1$H NMR spectra in 90:10 $H_2O:D_2O$ for J345 RNA only (**A**) and mixtures of RNA with lysozyme (**B**) and trypsin (**C**).

μM is interpreted to result from increasing non-condensate cluster formation (see cluster size distributions in *Figure 2—figure supplement 3* at [RP]=0 with increasing protein concentration). However, as in the confocal microscopy experiments, the low concentration of fluorescence-labeled RNA limits the detection of very small clusters where only one or zero of the RNA would be labeled. The FRET results are compared with theoretical predictions (*Figure 7B*) as detailed below.

We applied circular dichroism (CD) and nuclear magnetic resonance (NMR) spectroscopy with the goal of examining whether the proteins and RNA retain their folded states upon condensate formation. The CD spectra in *Figure 4—figure supplement 8* show that there is no substantial change in the shape of the spectrum of trypsin in the presence of the RNA from 225 to 250 nm, which would be expected if the protein had unfolded, as a random coil spectrum has essentially no ellipticity in this wavelength range and the spectrum. The key feature of the RNA spectrum, the broad peak at 250–290 nm, is also retained in the mixture. In fact, the spectrum of the trypsin-RNA mixture appears to be simply a linear combination of the spectra of each of the components measured separately.

NMR spectroscopic analysis of RNA-trypsin and RNA-lysozyme samples at PS-inducing concentrations focused on the structure of the RNA. We observed the characteristic $^1$H spectrum of a solution containing only J345 RNA that matches previously matched spectra for the same structure

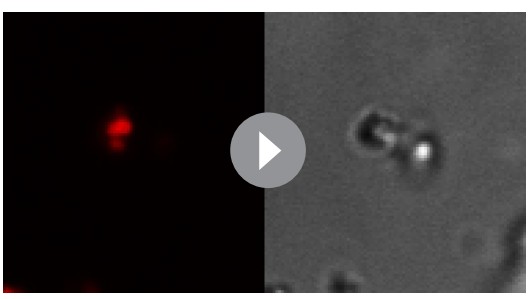

**Video 2.** Merging of trypsin-RNA liquid condensate droplets. Video of two representative examples of liquid droplet dynamics in trypsin-RNA mixtures with J345 RNA at 0.45 mM and proteins at 0.35 mM from confocal microscopy of fluorescent-labeled RNA (left) and corresponding bright-field imaging (right). Time evolution is accelerated 25x (i.e. the movies correspond to about 100 s in real time).

https://elifesciences.org/articles/64004#video2

(*Bonneau and Legault, 2014*; *Figure 4—figure supplement 9*). In the presence of proteins, the characteristic peaks were retained at the same positions, although with greatly attenuated intensities (*Figure 4—figure supplement 9*). This was interpreted to mean that only a fraction of RNA remained sufficiently dynamic to achieve rotational averaging via molecular tumbling. From comparing the signal-to-noise ratios, we estimate that about 80% of the RNA is not visible in the RNA-lysozyme sample and 90% is invisible in the RNA-trypsin sample. Since the majority of RNA is expected to be found in the condensates, this suggests that rotational diffusion of individual RNA molecules in the condensates is retarded significantly because the condensates themselves are too large (>100 nm) to tumble on time scales allowing NMR signals to be observed (<100 ns). Moreover, if one assumes that only RNA in the dilute phases remains visible in NMR spectroscopy, the experiments provide an estimate of the fraction of RNA in the dilute vs. condensed phases, that is 20:80 in the presence of lysozyme and 10:90 in the presence of the trypsin for the concentrations studied here. Unfortunately, that also implies that there is no information about the structure of RNA inside the condensates from these experiments.

## Phase separation of RNA and proteins described by simulations and theory

To compare with the experimental findings, we carried out CG simulations again with the model described above but for binary mixtures of spherical particles equivalent in size and charge to the experimentally studied systems, that is, J345 RNA (q=-46, r = 1.47 nm), myoglobin ($q$ = +2, $r$ = 1.64 nm), trypsin ($q$ = +6, $r$ = 1.81 nm), lysozyme ($q$ = +8, $r$ = 1.54 nm), lactate dehydrogenase (LDH; $q$ = +4, $r$ = 2.68 nm), alcohol dehydrogenase (ADH; $q$ = +8, $r$ = 2.79 nm), and bovine serum albumin (BSA; $q$ = −17, $r$ = 2.58 nm). We also tested a spherical particle equivalent to cytochrome C ($q$ = +11, $r$ = 1.45 nm) which was not studied experimentally because of heme absorption. We observed the formation of condensates at sufficiently high-salt concentrations. With $\kappa$ = 0.7 (about 20 mM salt), condensates formed with lysozyme, trypsin, LDH, and ADH, but not with cytochrome C, myoglobin, or BSA (*Figure 8*). Very similar results were also found with an alternative effective charge model (according to *Equation 6*) as shown in *Figure 8—figure supplement 1*.

The simulation results qualitatively match the experimental results in terms of which proteins promote PS. Moreover, the fraction of RNA in the dilute phase is higher with lysozyme than with trypsin (32% vs. 26–27% using *Equation 5* or *Equation 6* from averages over the last 100 μs) in qualitative agreement with the estimates from the NMR experiments. We note that an overall larger fraction of RNA is expected in the dilute phase in the simulations due to an excess concentration of RNA (0.439 mM) compared to the protein concentration (0.350 mM), whereas concentrations of RNA and protein were equal in

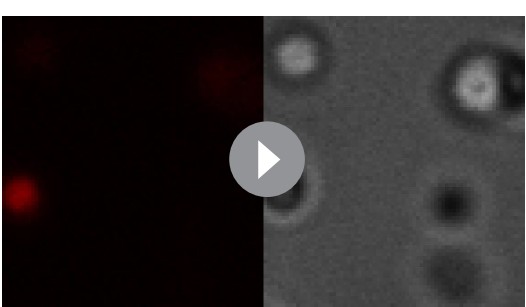

**Video 3.** Merging of trypsin-RNA liquid condensate droplets. Video of two representative examples of liquid droplet dynamics in trypsin-RNA mixtures with J345 RNA at 0.45 mM and proteins at 0.35 mM from confocal microscopy of fluorescent-labeled RNA (left) and corresponding bright-field imaging (right). Time evolution is accelerated 25x (i.e. the movies correspond to about 100 s in real time).

https://elifesciences.org/articles/64004#video3

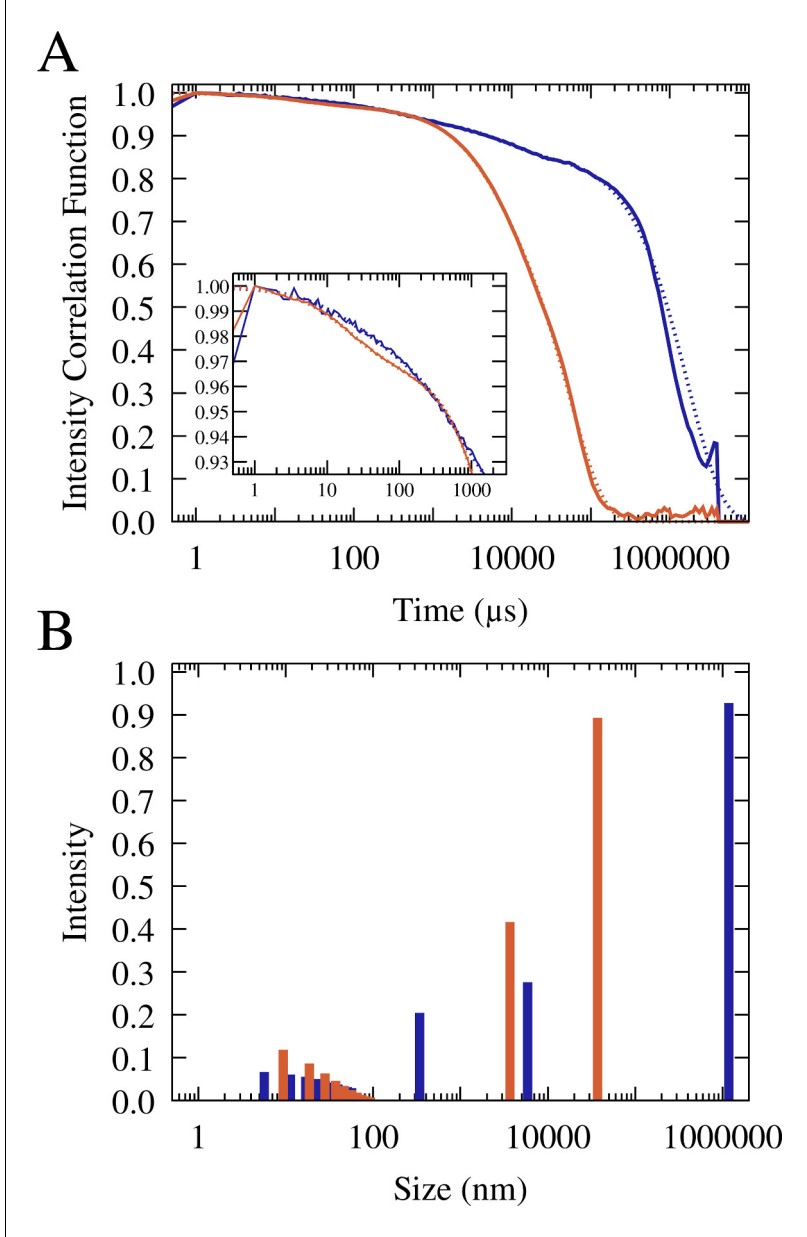

**Figure 5.** Normalized and averaged scattering intensity correlation functions from triplicate dynamic light scattering experiments of mixtures of 0.1 mM J345 RNA with 0.166 mM trypsin (orange) and 0.4 mM RNA with 0.675 mM lysozyme (blue) (A). Scattering intensity as a function of particle size from multi-exponential fits to the correlation functions (shown as dotted lines in A) for trypsin (orange) and lysozyme (blue) (B).

The online version of this article includes the following figure supplement(s) for figure 5:

**Figure supplement 1.** Dynamic light scattering results for RNA-trypsin condensates.

**Figure supplement 2.** Dynamic light scattering results for RNA-lysozyme condensates.

the NMR experiments (0.150 mM). However, the scale of the simulations is too small to directly compare the condensate sizes with the experimental size distributions.

To generate more extensive phase diagrams, a theoretical model was developed based on the CG simulations. Briefly, the model approximates the chemical potential for either RNA or proteins in condensed and dilute phases based on a decomposition into enthalpy and entropy: $\mu = \Delta h - T\Delta s$. The enthalpy is determined from convoluting the coarse-grained interaction potential $U(r)$ (*Equation 3*) with radial distribution functions $\hat{g}(r)$ of RNA-RNA, RNA-protein, and protein-protein

**Table 1.** Multi-exponential fits of dynamic light scattering correlation functions.

| System* | Clusters | | | Size 1 | | Size 2 | | Size 3 | | Size 4 | | $\chi^2$ |
|---|---|---|---|---|---|---|---|---|---|---|---|---|
| | $D_c$ (nm) | $a_c$ | $t_c$ | $D_1$ (nm) | $a_1$ | $D_2$ (nm) | $a_2$ | $D_3$ ($\mu m$) | $a_3$ | $D_4$ ($\mu m$) | $a_4$ | $*10^{-3}$ |
| Lysozyme #1 | 6.8 | 0.076 | 9.4 | 314.5 | 0.197 | 6061 | 0.309 | 1037.0 | 0.919 | | | 0.362 |
| Lysozyme #2 | 4.3 | 0.085 | 10.5 | 325.4 | 0.240 | 5416 | 0.300 | 730.4 | 0.908 | | | 0.91 |
| Lysozyme #3 | 4.0 | 0.045 | 21.7 | 270.0 | 0.160 | 2585 | 0.163 | 17.6 | 0.190 | 28,373.6 | 0.948 | 0.13 |
| Lysozyme avg. | 5.6 | 0.073 | 10.4 | 339.4 | 0.204 | 5848 | 0.275 | 1184.7 | 0.927 | | | 0.16 |
| Trypsin #1 | 7.6 | 0.129 | 5.0 | 2544 | 0.345 | 30,167 | 0.921 | | | | | 1.6 |
| Trypsin #2 | 2.7 | 0.051 | 186,625 | 2003 | 0.297 | 38,323 | 0.942 | | | | | 1.46 |
| Trypsin #3 | 2.4 | 0.055 | 106,796 | 5210 | 0.575 | 46,527 | 0.801 | | | | | 1.05 |
| Trypsin avg. | 9.3 | 0.162 | 3.2 | 3680 | 0.417 | 36,967 | 0.893 | | | | | 2.31 |

*All systems are mixtures between protein and J345 RNA.

interactions in the condensed and dilute phases extracted from CG simulations and scaled by particle densities $\rho$:

$$\Delta h = 2\pi\rho \int \hat{g}(r)U(r)r^2 dr \tag{1}$$

The entropy was estimated from the ratio of particle densities ρ between the entire system and either the dilute or condensed phase:

$$\Delta s = R\log\left(\frac{\rho_{total}}{\rho_{phase}}\right) \tag{2}$$

Solutions with respect to the concentrations of protein and RNA in dilute and condensed phases were determined numerically under the conditions that $\mu_{condensed} = \mu_{dilute}$ for either RNA, protein, or both, and that molecular volume packing fractions did not exceed maximum packing densities. Total free energies were then calculated, taking also into account mixing entropy contributions between RNA and protein particles. PS was predicted based on the solution with the lowest free energy.

The theoretical approach is essentially a variation of Voorn-Overbeek theory (**Overbeek and Voorn, 1957**) for spherical particles. While this theory has seen numerous applications, especially to polyelectrolyte fluids (**Priftis et al., 2014**; **Spruijt et al., 2010**), the specific model described here emphasizes an interaction potential that is parameterized based on atomistic simulations of biological macromolecules and that was further tuned to match experimental data. Therefore, the theory is expected to make predictions that are more relevant for globular biological macromolecules than previous studies.

In developing the theory, we found that using the alternative effective charge model according to **Equation 6** (**Figure 1—figure supplement 8**) results in better agreement between theory and experiment and therefore we used this model here. We also use a slightly different Debye-Hückel screening term, that is, κ = 1.17, which gave better agreement between theory and experiment.

Application of the theory predicts that PS should occur for a wide range of protein radii and charges as long as proteins are large enough and carry sufficiently positive charge (**Figure 9**). More specifically, radius/charge combination corresponding to lysozyme, trypsin, LDH, and ADH are predicted to lead to PS as in the experiments and CG simulations. The radius and charge corresponding to myoglobin is just outside the PS region (**Figure 9**) again consistent with the lack of PS in the experiment and simulations. The theory also predicts PS for cytochrome C, for which PS was not seen in the simulations.

The theory reproduces an expected temperature dependence of PS with protein-dependent critical maximal temperatures (**Figure 8—figure supplement 2**). The electrostatic nature of PS also suggests that changes in salt concentrations would affect the findings and the results are indeed sensitive to the value of κ. However, the theoretical treatment is too limited due to the mean-field

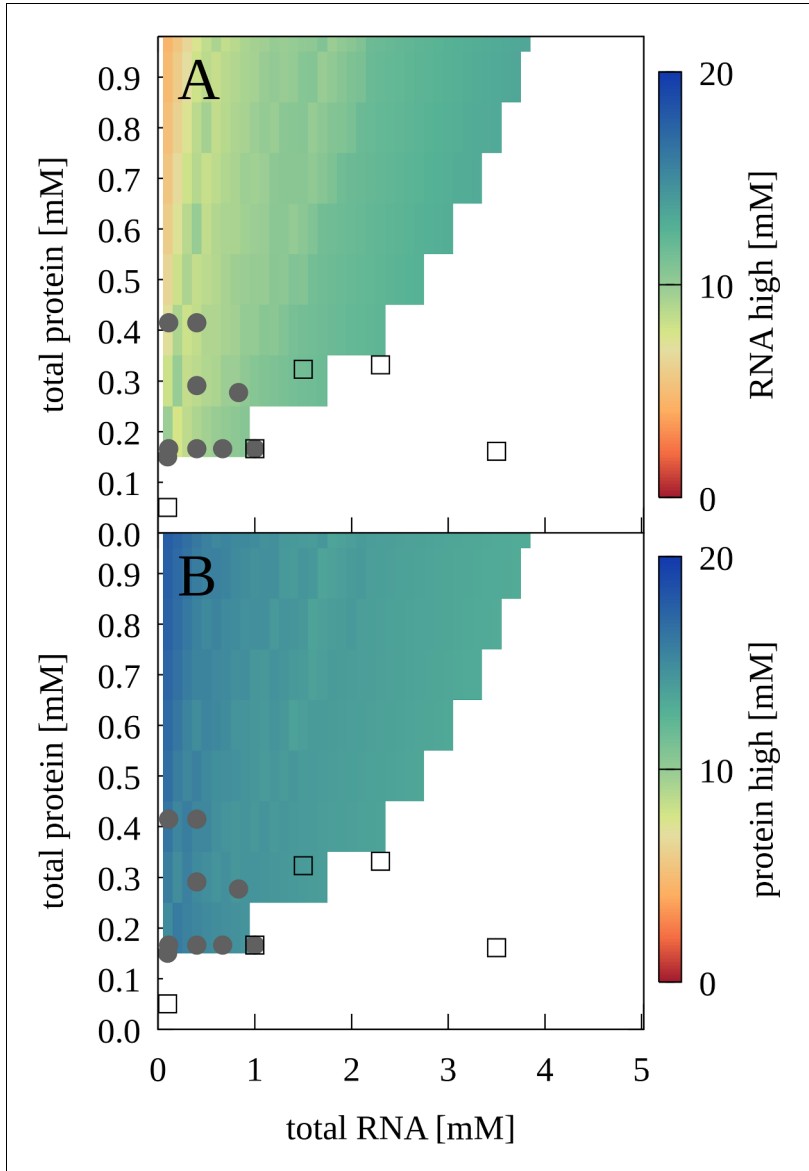

**Figure 6.** Phase separation for mixtures of J345 RNA and trypsin as a function of total protein and RNA concentrations from experiment and theory. Grey filled circles indicate concentrations for which phase separation was observed experimentally based on confocal microscopy; empty squares indicate concentrations for which microscopy imaging did not show phase separation. Colors indicate predicted concentrations from theory for RNA (A) and proteins (B) in the condensed phases. No phase separation is predicted for white areas.

The online version of this article includes the following figure supplement(s) for figure 6:

**Figure supplement 1.** Confocal microscopy of labeled J345 RNA for mixtures between J345 RNA at 0.1 mM and trypsin at 0.05 mM (**A**) and at 0.15 mM (**B**).

nature of the Debye-Hückel formalism to make meaningful predictions of salt effects. More specifically, the model is only valid for low ionic strengths and ignores entropic consequences of ion partitioning between condensed and dilute phases that are an important contribution to PS in complex coacervates (*Vis et al., 2015*).

   Using the theory, we constructed concentration-dependent phase diagrams that can be compared with experiment. *Figure 6* shows the prediction of the two-phase region for RNA-trypsin in good agreement with the experimental data. *Figure 9—figure supplements 1–6* show the phase diagrams for all proteins studied here over a wider range of concentrations. All phase diagrams

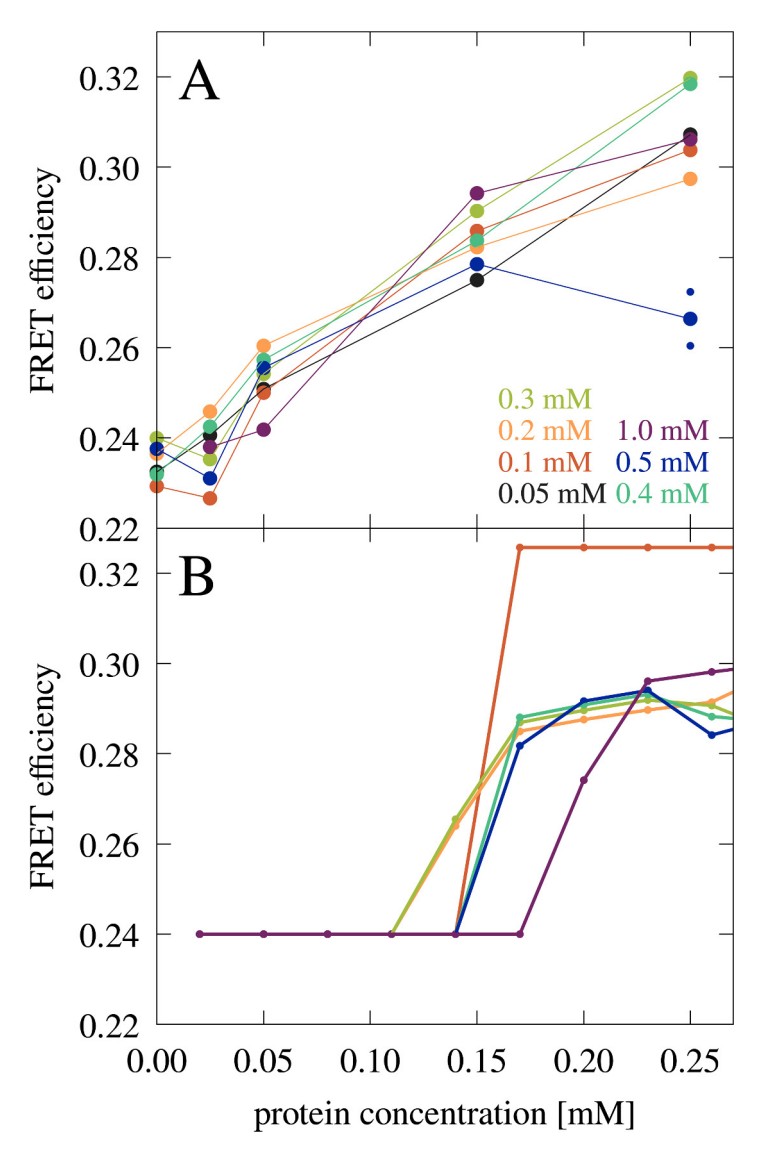

**Figure 7.** FRET efficiency in mixtures of J345 RNA with trypsin as a function of protein concentration at different RNA concentrations (as indicated by color). The average of two measurements is shown for 0.25 mM protein and 0.5 mM RNA concentrations with smaller points indicating individual measurements. (**A**) FRET efficiency estimated from the fraction of RNA in the condensed phase from theory (**B**). In every measurement, the concentration of Cy3- and Cy5-labeled RNA is constant, 8 µM and 42 µM, respectively.

exhibit reentrant behavior with minimal and maximal protein and RNA concentrations as expected for complex coacervates. It should be noted, though, that the full range of concentrations cannot be realized in practice for all systems due to limited solubilities.

Predictions from the theory also allowed a quantitative interpretation of the FRET experiments. Using the predicted fraction of RNA in the condensates for the RNA-trypsin mixtures at different RNA and protein concentrations (*Figure 9—figure supplement 7*), FRET efficiencies were estimated (*Figure 7B*). The theoretical predictions qualitatively reproduce the experimental data with an onset of increased FRET efficiencies due to condensation. Moreover, the gradual increase in FRET efficiencies after condensates form is predicted from a growing number of RNA in the condensed phase as protein concentration increases.

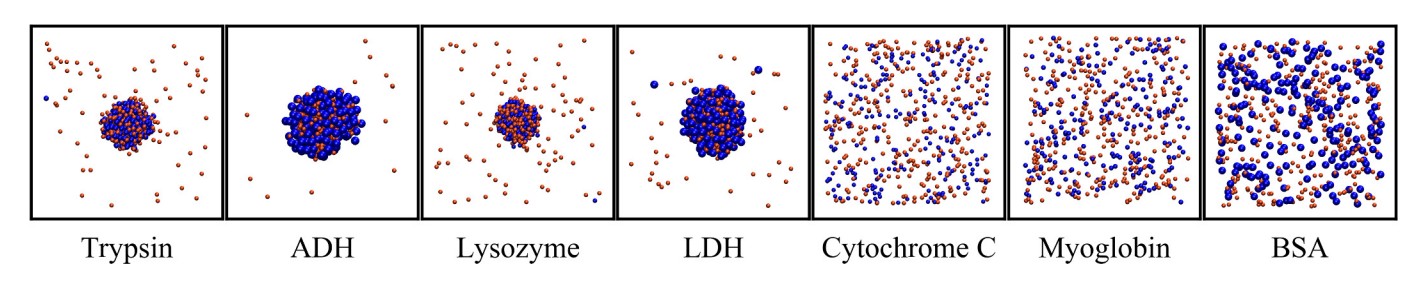

**Figure 8.** Snapshots after 1 ms for binary RNA-protein mixtures at T = 298K, with κ = 0.7 and using effective charges according to *Equation 5*. [RNA] =0.493 mM and [protein]=0.350 mM. Orange and blue spheres show RNA and proteins, according to size. Concentrations inside the condensates were [RNA:lysozyme]=20.2:20.2 mM; [RNA:trypsin]=16.5:15.2 mM; [RNA:LDH]=9.6:7.2 mM; [RNA:ADH]=9.5:6.7 mM.

The online version of this article includes the following figure supplement(s) for figure 8:

**Figure supplement 1.** Snapshots from CG simulations after 1 ms for binary RNA-protein mixtures at T = 298K, with κ = 0.75 using *Equation 6* to obtain effective charges.

**Figure supplement 2.** Concentrations of RNA (**A**) and proteins (**B**) in dilute and condensed phases as a function of temperature with κ = 1.17.

**Figure supplement 3.** Charge distribution on protein surfaces based on amino acid residue types (top; basic: blue, acidic: red, polar: green, hydrophobic: white) and electrostatic potentials calculated via a Poisson-Boltzmann continuum model (bottom) with coloring according to the sign of the potential (positive: blue, negative: red).

## Discussion

This study presents a general view on charge-driven biomolecular PS supported by simulation, theory, and experiments. More specifically, we report a potential for PS between negatively charged RNA and positively charged proteins without requiring polymer-character of either component or specific binding interactions. Our simulations and the theoretical model are based on isotropic spheres, whereas experimental validation is based on a compact, approximately globular RNA and a variety of globular proteins that are not known to specifically interact with RNA. This implies that PS may be a very general phenomenon in biological cells depending on the concentrations, charge, and size distribution of available nucleic acid and protein components. In fact, our simulations of a bacterial cytoplasm provide examples of separately forming tRNA-protein and ribosome-protein condensates involving a variety of proteins in a cytoplasmic environment. Separate condensates of nucleic acids with different charge and size could have important implications for the role of PS in vivo.

The idea of strong complementary electrostatic interactions playing a major role in PS via complex coacervate formation is well-established for a variety of different molecules (*Sawyer et al., 2019*; *de Kruif et al., 2004*; *Fay and Anderson, 2018*; *Cummings and Obermeyer, 2018*; *Michaeli et al., 1957*; *Mattison et al., 1995*) and also for PS involving biomolecules (*Ghosh et al., 2019*). While almost all the LLPS studies to-date involve polymers and in particular IDPs (*Dignon et al., 2018a*), there are also examples in the literature that discuss PS involving folded proteins (*Sanders et al., 2020*; *Aumiller et al., 2016*; *Banani et al., 2016*; *Li et al., 2012*; *Banjade and Rosen, 2014*; *Conicella et al., 2020*). In most of those cases, the ability to form condensates is generally ascribed to specific multi-valent interactions and evidence for a more generic electrostatic-only mechanism are only just beginning to emerge (*Cummings and Obermeyer, 2018*; *Sanders et al., 2020*). The results presented here provide evidence for a more general principle that does not require flexible polymers, specific interaction sites, or specific secondary structures (*Conicella et al., 2020*). The central principle is simply electrostatic complementarity at the molecular level, but a more generalized concept of multi-valency is implicitly assumed. Isotropic spheres without any directional preference for interactions are in fact maximally multi-valent, limited only by the excluded-volume interactions between the binding partners. On the other hand, globular proteins with basic amino acids distributed widely across their surface and diffuse positive electrostatic potentials over most of the molecular surface (*Figure 8—figure supplement 3*) are effectively poly-valent particles with respect to interactions with nucleic acids. The key insight from this study is that proteins not known to interact specifically with nucleic acids under dilute conditions may form condensates with

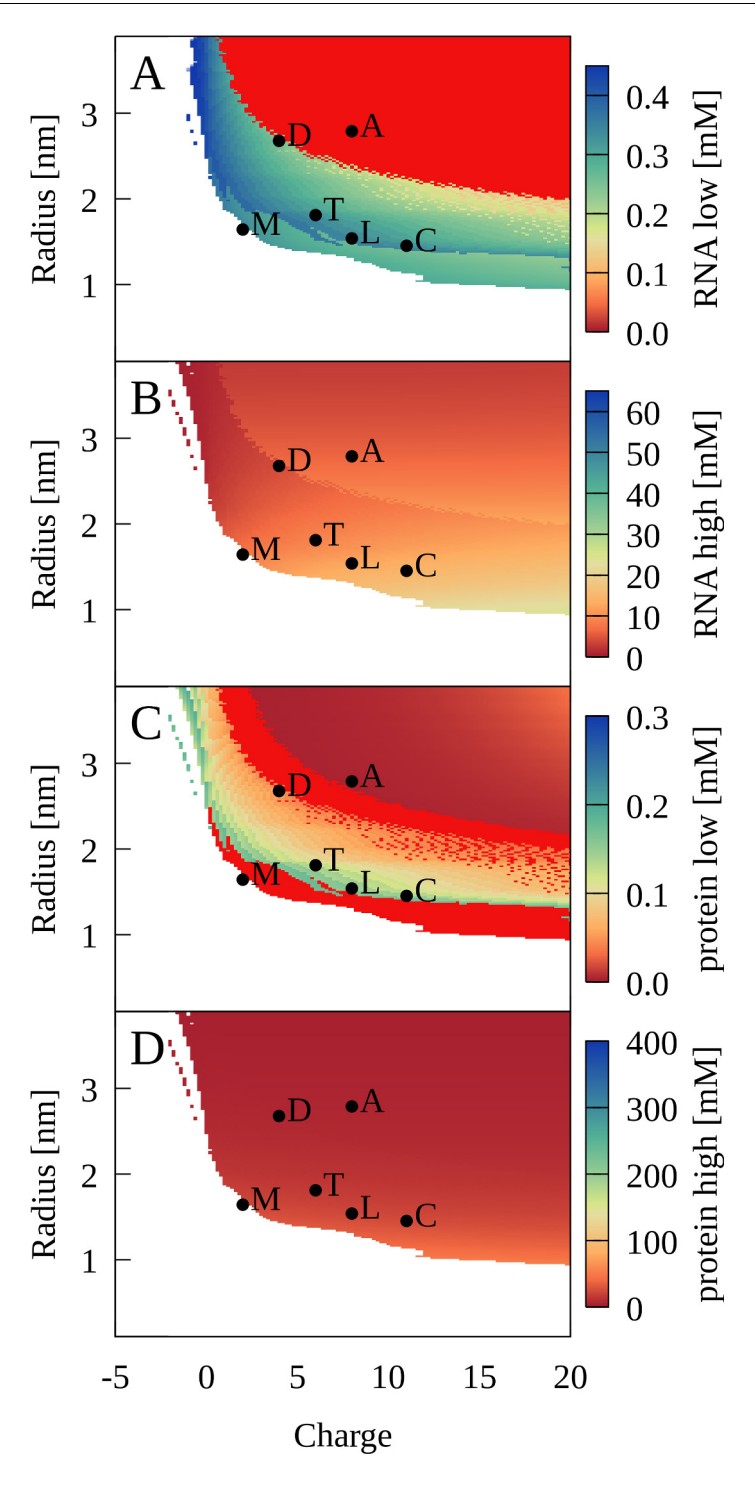

**Figure 9.** Phase separation for binary RNA-protein mixtures as a function of protein charge and radius from theory. Colors show [RNA] (**A**, **B**) and [protein] (**C**, **D**) in dilute (**A**, **C**) and condensed (**B**, **D**) phases. Red indicates zero concentration. [RNA]=0.45 mM, [protein]=0.35 mM, κ = 1.17, and T = 298 K. Corresponding properties for proteins are denoted as follows: myoglobin (M); trypsin (T); lysozyme (L); cytochrome C (C); LDH (D); ADH (A). The online version of this article includes the following figure supplement(s) for figure 9:

**Figure supplement 1.** Phase separation for mixtures between RNA and alcohol dehydrogenase as a function of total protein and RNA concentrations.

*Figure 9 continued on next page*

*Figure 9 continued*

**Figure supplement 2.** Phase separation for mixtures between RNA and lactate dehydrogenase as a function of total protein and RNA concentrations.

**Figure supplement 3.** Phase separation for mixtures between RNA and lysozyme as a function of total protein and RNA concentrations.

**Figure supplement 4.** Phase separation for mixtures between RNA and trypsin as a function of total protein and RNA concentrations.

**Figure supplement 5.** Phase separation for mixtures between RNA and cytochrome C as a function of total protein and RNA concentrations.

**Figure supplement 6.** Phase separation for mixtures between RNA and myoglobin as a function of total protein and RNA concentrations.

**Figure supplement 7.** Fraction of RNA (**A**) and protein (**B**) in the condensed phases predicted by the theory model for trypsin as a function of protein concentration at different total RNA concentrations.

---

nucleic acids, if the proteins are present at sufficient amounts, simply based on a principle of generic poly-valency and an overall charge attraction.

Our study suggests that size and charge are essential determinants of PS between RNA and proteins. Favorable condensates require optimal packing and a balance of attractive and repulsive interactions between oppositely charged RNA and protein particles. *Figure 1—figure supplement 9* shows a snapshot from the cytoplasmic system illustrating how such packing may be achieved. The optimal balance depends on the size of the RNA particles: Larger proteins are required for the smaller RNA molecules to phase separate, whereas smaller proteins allow the larger ribosomal particles to phase-separate (*Figure 1—figure supplement 5*). This can be seen more clearly in the five-component model system, where a relatively modest reduction in the radius of the larger positively charged particle leads to a loss of close tRNA contacts (*Figure 1—figure supplement 10*), therefore preventing condensate formation. The theoretical model for binary RNA-protein mixtures also predicts a minimum protein radius for PS, at least at lower charges (*Figure 9*). Myoglobin is outside the predicted range and although it has a net-positive charge, PS was not observed in the experiment at protein concentrations below the RNA concentrations (*Figure 4*) consistent with the theory. The sensitivity to matching size and charge between the RNA and proteins suggests at least a partial explanation for the observation of separate condensates for tRNA and RP in the simulations of the cytoplasmic model systems.

The total concentration of the protein is another determinant for PS. Simulations and theory predict minimum protein concentrations depending on the protein charge and size around 0.05 mM or more (*Figure 9—figure supplements 1–6*). For trypsin, this was validated experimentally via microscopy and FRET spectroscopy (*Figures 6* and *7*). While many cellular proteins may not be present individually at such high concentrations, our cytoplasmic model shows that a heterogeneous mixture of similar-sized and similar-charged proteins may promote PS equally well. At the lower end, the RNA concentration appears to be a less critical factor for observing PS, although a larger amount of RNA allows more numerous and larger condensates to form, assuming that there is enough protein available, at least until reaching a critical RNA concentration beyond which PS is not favorable anymore. In binary mixtures, this is simply a question of the total protein concentration. In the heterogeneous cytoplasmic model, we found competition for the larger positively charged proteins by the ribosomes forming their condensates to be another factor affecting tRNA condensate formation that would need to be considered in cellular environments (*Figure 2—figure supplement 7*).

Since electrostatics is a major driving force of the PS described here, changes in salt concentration are expected to alter the tendency for PS. The theory applied here is not well-suited to examine variations in the salt concentration. At the same time, there is only a limited range of decreased salt conditions that can be applied before either the RNA or the protein structures become destabilized. Therefore, we could not yet develop an accurate quantitative understanding of how salt effects may affect charge-driven phase separation. This topic will have to be deferred to future studies.

A significant interest in PS in biology is related to liquid-state condensates. Such condensates would maintain the dynamics that is necessary for many biological processes as opposed to dynamically retarded gels or amorphous clusters. The simulations suggest that the condensates retain significant dynamics based on calculated self-diffusion rates, although there are serious limitations on

diffusion estimates from CG simulations, especially in the absence of hydrodynamic interactions (*Ando and Skolnick, 2010*). In experiment, we find evidence of liquid-like behavior for condensates formed in RNA-trypsin mixtures, but the dynamic properties of RNA or proteins in other RNA-protein condensates are less clear. As another data point, the NMR spectroscopy results also suggest significant retardation of diffusional dynamics inside the condensates.

Although there are some limitations in the current study that will need to be revisited in future studies to gain a more detailed understanding of the more universal PS between RNA and proteins described here, the main advantage of the CG models and theory is that its simplicity allowed us to explore the large spatial scales and long-time scales that can predict phase behavior on experimentally accessible scales. The CG models were parameterized based on high-resolution atomistic simulations of concentrated protein solutions, these models lack all but the most basic features of biological macromolecules. Increased levels of realism could be achieved without too much additional computational cost via patchy particles (*Nguemaha and Zhou, 2018*), whereas higher-resolution in the form of residue-based coarse-graining (*Dignon et al., 2018a*) to explore the effects of shape anisotropy and inhomogeneous charge distributions across RNA and protein surfaces is in principle attainable but computationally much more demanding.

The cytoplasmic model described here is a first step toward modeling biologically relevant environments, but leaves out DNA, membranes, and other cellular structures such as the cytoskeleton. The CG version of the cytoplasmic model furthermore neglects metabolites, whereas the representation of macromolecules as spheres is clearly an oversimplification, especially for more flexible and irregularly shaped molecules such as mRNAs or proteins with significant intrinsic disorder or internal dynamics. Future studies will aim to include the missing factors to examine how important such additional details are for modulating phase separation processes in vivo.

Finally, we expect that further insights could be gained from atomistic simulations of RNA-protein clusters initiated from configurations in the CG simulations to better understand the detailed molecular interactions stabilizing the condensates. On the experimental side, we only focused on RNA without visualizing protein condensation. Moreover, there is a need to follow up on this work with in vivo studies to establish how ubiquitous the condensates described here are under cellular conditions.

## Conclusions

We report phase separation of RNA and proteins based on a universal principle of charge complementarity that does not require polymers or multi-valency via specific interactions. The results are supported by coarse-grained simulations, theory, and experimental validation via microscopy, FRET, and NMR spectroscopy as well as DLS experiments. Condensate formation depends on concentration, size, and charge of the proteins but appears to be possible for typical RNA and common proteins. Simulation results, furthermore, suggest that such phase separation may occur in heterogenous cellular environment, not just between tRNA and cellular proteins but also, in separate condensates, between ribosomes and proteins. Further computational and experimental studies are needed to gain more detailed insights into the exact molecular nature of the condensates described here.

The larger implication of the work presented here is that charge-driven phase separation appears to be a broad phenomenon in biology, particularly because intrinsically disordered proteins and disordered RNA are not required. As a result, cellular cytoplasms could be phase-separated extensively. The observation that tRNA could condense and co-locate near ribosomes suggests a mechanism in which the rate of protein translation is increased because the diffusional wait time for the correct tRNA arriving at the ribosome is decreased. It remains to be explored through in vivo experiments how widely charge-driven phase separation may present itself in cellular environments and what additional factors may modulate it.

## Materials and methods

### Coarse-grained model

CG simulations were run using a modified version of a previously introduced colloid-type spherical model (*Mani et al., 2014*). In this model, pair interactions consist of a short-range 10–5 Lennard-Jones potential and a long-range Debye Hückel potential according to:

$$U(r_{ij}) = 4\varepsilon\left(\left(\frac{\sigma_{ij}}{r_{ij}}\right)^{10} - \left(\frac{\sigma_{ij}}{r_{ij}}\right)^{5}\right) + \frac{(A_{ij}+A_0)\kappa\sigma_{ij}}{r_{ij}}e^{-\frac{r_{ij}}{\kappa\sigma_{ij}}} \tag{3}$$

where $r_{ij}$ is the inter-particle distance, $\sigma_{ij}$ is the distance between particles at which the potential is zero, $\varepsilon$ is the strength of short-range attraction, $A_{ij}+A_0$ describes attractive or repulsive long-range interactions, and $\kappa\sigma_{ij}$ is the Debye-Hückel screening length. Only $A_{ij}$ and $\sigma_{ij}$ vary between different particles according to charge and size.

The model was initially parameterized from previously published all-atom simulations of homogeneous mixtures of chicken villin headpiece ('villin') (*Nawrocki et al., 2017*) and subsequently validated with heterogeneous mixtures of protein G, villin, and ubiquitin (*Nawrocki et al., 2019a*) as summarized in *Table 2*.

A common value of $\varepsilon$ = 4.0 kJ/mol was used for all particles in the short-range 10–5 Lennard-Jones potential. Particle size was taken into account by first determining the radii $r_i$ of spheres with equivalent volumes to the atomistic molecular volumes of a given macromolecule or complex (see *Supplementary file 1* for molecules in the cytoplasmic model system). Lennard-Jones parameters $\sigma_i$ were obtained from the radii $r_i$ according to:

$$\sigma_{\mathrm{i}} = 2^{-\frac{1}{6}}\cdot r_i \tag{4}$$

Pairwise parameters $\sigma_{ij}$ were calculated as $\sigma_{ij} = \sigma_i + \sigma_j$.

In the long-range Debye-Hückel type potential, a common value of $A_0$ = 3.0 kJ/mol was used to reflect effective repulsion between charge-neutral, but still polar molecules due to solvation effects. Net charges led to additional repulsive or attractive contributions.

The nominal net charge of a given molecule was converted to effective charges to account for counterion condensation around highly charged macromolecules (*Manning, 1978*). We distinguish here effectively bound ions that lead to an effectively reduced charge vs. ions that remain mobile in solution and give rise to Debye screening as described below. Generally, the effective charge remains close to nominal charges for small charges, but for highly charged molecules, in particular negatively charged nucleic acids and nucleic acid complexes such as the ribosome, the effective charge is reduced significantly (*Diehl and Levin, 2004*; *Wishnia and Boussert, 1977*; *Trylska et al., 2004*). Charge neutralization is more pronounced with divalent ions such as $Mg^{2+}$ vs. monovalent ions such as $Na^+$ or $K^+$ (*Diehl and Levin, 2004*; *Templeton and Elber, 2018*; *Yoo and Aksimentiev, 2012*). But the amount of $Mg^{2+}$ ions in biological systems is limited and typically not high enough to neutralize the charge of all the nucleic acids so that additional charge neutralization by monovalent ions remains a significant factor (*Akanuma et al., 2014*).

Here, we propose the following two expressions to obtain effective charges:

**Table 2.** Simulation systems for coarse-grained model validation.

| System | Villin | | | Protein G | | | Ubiquitin | | | Box (nm) |
|---|---|---|---|---|---|---|---|---|---|---|
| Volume percentage | G/L | mM | $N_p$* | G/L | mM | $N_p$* | G/L | mM | $N_p$* | |
| 5% | 9.7 | 2.3 | 5 | 14.3 | 2.3 | 5 | 19.8 | 2.3 | 5 | 15.3 |
| 10% | 19.0 | 4.5 | 10 | 28.2 | 4.5 | 10 | 39.0 | 4.5 | 10 | 15.4 |
| 30% | 57.9 | 13.8 | 30 | 85.7 | 13.8 | 30 | 118.6 | 13.8 | 30 | 10.6 |

*Number of proteins.

$$q_{\text{eff},1} = \text{sign}(q) \cdot 20 \cdot \log\left(\frac{|q|}{20} + 1\right) \tag{5}$$

$$q_{\text{eff},2} = \text{sign}(q) \cdot 0.6\sqrt{|q|} \cdot \log\left(\frac{|q|}{2} + 1\right) \tag{6}$$

Both empirical formulae give effective charges close to nominal charges for molecules with small charges and highly reduced charges for molecules with large formal charges (*Figure 1—figure supplement 8*). For a DNA molecule with a nominal charge of −45, atomistic MD simulations suggest effective charges of −10 to −20 under the assumption that ions within 1 nm from the solute surface are effectively bound (*Templeton and Elber, 2018*; *Yoo and Aksimentiev, 2012*); at the other end, effective charges between −100 and −800 are estimated for ribosomal particle with a nominal charge of about −4000 based on colloid models (*Diehl and Levin, 2004*) or electrostatic potential calculations (*Trylska et al., 2004*), assuming a mixture of divalent and monovalent ions is involved in neutralization. *Equations 5 and 6* are both consistent with these estimates. *Equation 5* was used initially and screens smaller charges less and larger charges more strongly compared to *Equation 6* which was adopted after adjusting the theory to better match experimental results. Neither expression considers ionic concentration as counterion condensation does not depend strongly on concentration (*Yoo and Aksimentiev, 2012*). Moreover, negatively and positively charged solutes are treated in the same manner even though the binding strength of biological anions (Cl⁻) and cations (K⁺, Na⁺, Mg²⁺) to oppositely charged macromolecules may be asymmetric. However, since highly positively charged macromolecules are uncommon, this assumption may not have significant consequences for the systems studied here.

The effective charges calculated either via *Equation 5* or *Equation 6* were then converted to $A_i$ values:

$$A_i = \text{sign}(q_i)\sqrt{\frac{3}{4}q_{i,\text{eff}}} \tag{7}$$

Pairwise values were determined as $A_{ij}=A_i{}^*A_j$ and the factor ¾ was determined by parameterization against the atomistic MD simulations.

The Debye screening length in *Equation 3* is $\kappa\sigma_{ij}$, that is, it depends on particle size as in the original model by *Mani et al., 2014* in order to better model screening interactions between particles of very different sizes with screened charges that are mostly near the surface. This complicates interpretation of $\kappa$ in terms of specific salt concentrations. However, as an illustration one may consider a typical smaller protein or RNA with $\sigma_{ii}$ of 3 nm where $\kappa$ = 0.5, 1.0, and 1.5 would correspond to monovalent ion concentrations of 40, 10, and 5 mM, respectively. Note, that these ion concentrations reflect excess ion concentrations after subtracting condensed counterions as those are accounted for in the effective charges according to *Equation 5* or 6. Therefore, total ion concentrations in experiment corresponding to a given value of $\kappa$ in our model should be significantly higher by factors of 2–10 depending on the charges of the considered macromolecules.

## Coarse-grained molecular dynamics simulations

MD simulations of the CG model were run up to 1 ms using OpenMM (*Eastman et al., 2017*) on GPU machines. The interaction potential from *Equation 3* was implemented as a custom nonbonded interaction potential via OpenMM's Python interface. A Langevin thermostat was applied with a temperature of 298 K unless noted otherwise and a friction coefficient of 1 ps⁻¹. As a result, the simulations described here reflect stochastic dynamics of our CG model. A value of $\kappa$ = 1.5 was used to describe salt screening unless noted otherwise. The timestep for the simulations was set to 1 ps. Frames were saved every 1 ns for simulations of the 100 nm cytoplasm model, every 10 ns for the concentrated protein simulations used for parameterization, and every 100 ns for all other systems. The pairwise potential in *Equation 3* was evaluated with a cutoff 49.5 nm. A switching function was applied to be effective at 49 nm. In total, about 270 ms of combined simulation time was run for all systems described here. The total computational cost for these simulations was around 350 GPU days based on timing on a single NVIDIA GeForce GTX 1080 Ti GPU card.

For validation, CG simulations of the systems with the same concentrations as in the atomistic simulations were performed for 100 µs. The CG simulations compared favorably with the atomistic simulations based on pairwise radial distribution functions and cluster size distribution (*Figure 1— figure supplement 11*).

### Bacterial cytoplasm model

We constructed a coarse-grained model of *Mycoplasma genitalium* cytoplasm based on our previously established atomistic model (*Yu et al., 2016*; *Feig et al., 2015*). All the macromolecules and complexes were converted to single spherical particles where the particle center initially coincided with the center of mass of the molecules in the atomistic model. Sphere radii were determined as described above based on equivalent volumes, and effective charges were determined from nominal charges according to *Equation 5* or *Equation 6*. A list of all particles with their size, charge, effective charge and concentration is given in *Supplementary file 1*. The initial system is a cubic box with a size of 100 nm. Additional systems were generated with 200 and 300 nm box sizes by replicating the initial system accordingly. MD simulations were run up to 1 ms as described above.

### Five-component model systems

A representative model of the cytoplasmic system consisted of five components, with an effective charge and volume fraction matching the values in the cytoplasmic system. The components consist of tRNA, ribosome particles (RP), positively charged proteins with small ($POS_S$) and large ($POS_L$) sizes and charges and neutral crowders (CRW) (*Supplementary file 2*). tRNA and RP have the same size and charge as in the full cytoplasmic system. The RP concentration includes RP, that is, complete ribosomes, in the cytoplasmic model as well as additional numbers of ribosomal fragments RR23, R50P RR16 and R30P (*Supplementary file 1*). The tRNA concentration was adjusted to include all particles with a nominal charge between −100 and −25, except for GroEL, which has a very large size and was not found as part of the tRNA condensates in the cytoplasmic simulations. Concentrations of the positively charged proteins were adjusted to keep the total effective charge of the system close to the cytoplasmic model. The system components were then varied to achieve different concentrations of RP and positively charged particles (*Supplementary file 2*). Simulations of the five-component system were performed as described above over 1 ms using only effective charges calculated via *Equation 5*.

### Two-component model systems

Two-component RNA-protein systems were simulated with the same CG model as described above for 1 ms to make predictions for experimentally testable systems. Effective charges were calculated either via *Equation 5* or *Equation 6*. RNA particles were modeled after the 47-nucleotide J345 Varkud satellite ribozyme RNA, that folds into an approximately globular shape (*Bonneau and Legault, 2014*) with $r_{RNA}$ = 1.47 nm and $q_{RNA}$ = −46. Proteins were considered with the following charges and radii: myoglobin (+2, 1.64 nm), trypsin (+6, 1.81 nm), lysozyme (+8, 1.54 nm), cytochrome C (+11, 1.45 nm), lactate dehydrogenase (+4, 2.68 nm), alcohol dehydrogenase (+8, 2.79 nm), and bovine serum albumin (−17, 2.58 nm).

### MD simulation analysis

Analysis of the CG simulations was performed for the simulation time between 500 µs to 1 ms unless stated otherwise using in-house code in conjunction with the MMTSB Tool Set (*Feig et al., 2004*).

#### Cluster analysis

We previously analyzed macromolecular clustering using specific distance cutoffs that were suitable for capturing direct molecular interactions leading to transient clusters (*Nawrocki et al., 2017*; *Nawrocki et al., 2019a*; *Nawrocki et al., 2019b*). From those studies, we arrived at a definition of clusters based on contacts where center of mass distances between spherical particles are less than $\sigma_{ij}$ + 0.7 nm. $\sigma_{ij}$ is the pair-wise Lennard-Jones parameters in *Equation 1* defined as described above in *Equation 4*. This criterion was applied to all pairs of particles, of same or different type, and connected graphs were generated from the pairs determined to be in contact. All particles within such a graph were then considered to be part of one cluster.

We initially applied this criterion here as well in a slightly modified version where we only considered contacts based on tRNA-protein and RP-protein pairs in order to be able to separately analyze tRNA and RP clustering in the same system. GroEL-protein pairs were also included when analyzing RP clusters since they were found to associate on the surface of the RP-rich condensates. We found that the $\sigma_{ij}$ + 0.7 nm contact criterion underestimated cluster sizes when visually inspecting condensed states (*Figure 1—figure supplement 12*). This may not be surprising since macromolecules in condensates are not necessarily in direct contact with other molecules while direct interactions are the essential feature of the transient molecular clusters described by us previously. From inspecting radial distribution functions for interactions between tRNA and $POS_L$ and $POS_S$ particles in the five-component system at different concentrations, we found that an increased cutoff of $\sigma_{ij}$ + 2.2 nm would include all the contacts within the first peak (*Figure 2—figure supplement 9*).

We further validated whether this criterion is more generally applicable to the cytoplasmic system by comparing with results from geometry-based scale-free hierarchical clustering. We applied such an algorithm to just tRNA particles during the last 100 µs of the simulation of the cytoplasmic systems so that clusters could be defined without having to invoke any contact-based criteria and without having to define clusters via interactions with other system components. We used the hierarchical clustering method implemented in the MMTSB Tool Set (*Feig et al., 2004*), but with a more recently established criterion for determining the optimal number of clusters (*Zhou et al., 2017*). This approach gave fluctuating cluster sizes between 180 and 260 tRNA molecules with a peak near 240 molecules (*Figure 1—figure supplement 13*). Clusters based on the $\sigma_{ij}$ + 2.2 nm distance cutoff for tRNA-protein pairs resulted in a narrower distribution but with a peak at the same number of molecules, whereas shorter cutoffs gave significantly smaller clusters. The broader variation in cluster sizes from the geometrical clustering reflects in part a lack of robustness in estimating optimal cluster sizes from scale-free hierarchical clustering (*Zhou et al., 2017*), and this is also the reason for why we used the contact-based criterion here instead of hierarchical geometrical clustering for determining tRNA and RP clusters.

## Diffusion analysis

Translational diffusion ($D_{tr}$) was calculated for each molecule in the cytoplasmic system from the mean square displacement (MSD) of molecules between time $t$ and $(t+\tau)$ for a given lag time $\tau$. Diffusion coefficients were then obtained from linear fits to MSD($\tau$) vs. $\tau$ (*Figure 1—figure supplement 6*).

$$D_{tr} = \frac{\text{MSD}(\tau)}{6\tau} \tag{8}$$

The first and last 1 µs of the cytoplasmic simulations were resampled so that conformations could be saved with a 1-ns interval. This allowed the analysis of all molecules in the dispersed and condensed states at the beginning and end of the trajectory and a comparison with previously published diffusion rates of macromolecules in the same system simulated in atomistic detail during similar time scales (*Yu et al., 2016*). In this case, the slope of MSD($\tau$) was fitted up until $\tau$ = 20 ns. Diffusion coefficients were calculated separately for molecules inside the tRNA and RP condensates as well as for molecules in the dilute phase. Molecules were considered to be part of a condensate if they remained part of the condensate during the entire lag time $\tau$.

For the five-component model system, diffusion was analyzed based on the last 100 µs of the simulations based on snapshots saved with a 100-ns interval and determining the slope of MSD($\tau$) up until $\tau$ = 2 µs.

## Phase separation analysis

In order to determine critical temperatures, CG simulations were performed at temperatures ranging from 300 to 500 K in 10 K increments using the Langevin thermostat. The critical temperatures and concentration were obtained by fitting the temperature to the coexisting volume fractions using the following formulas (*Guggenheim, 1945*):

$$\phi_H - \phi_L = A(T_c - T)^{0.32} \tag{9}$$

$$\frac{1}{2}(\phi_H + \phi_L) = \phi_C + B(T - T_c) \tag{10}$$

where $\varphi_H$ and $\varphi_L$ are the volume fractions of tRNA inside and outside of the clusters respectively, $T$ is the temperature, $T_c$ is the critical temperature and $\varphi_c$ is the critical volume fraction. This calculation was done for the model system simulations at different RP and $POS_L$ concentrations (*Supplementary file 2*).

## Analytical theory describing condensation between RNA and proteins

An analytical model was constructed to reproduce the phase behavior seen in the simulations and allow a wider range of parameters to be explored. The analysis focuses on a two-component system consisting of a mixture of negatively charged particles $R$, equivalent to the RNA in the simulations, and particles $P$, equivalent to proteins, typically with a positive charge. The particles have charges $q_R$, $q_P$ and radii $r_R$, $r_P$. We consider a system of volume V in which R and P particles are present in total concentrations of $c_R$ and $c_P$. However, we do not include any finite-size effects and therefore the following analysis is scale-independent.

We assume that a phase-separated state is formed with a high-density condensate of volume $V_c$ and a low-density dilute phase of volume $V_d = V - V_c$, that is, there is no change in the total system volume upon phase separation. The concentrations of R and P particles in the dilute and condensed phases are denoted as $c_{R,d}$, $c_{P,d}$, $c_{R,c}$, and $c_{P,c}$. From the concentrations, number densities $\rho_{R,d}$, $\rho_{P,d}$, $\rho_{R,c}$, and $\rho_{P,c}$ for R and P particles in the dilute and condensed phases are obtained according to $\rho = \frac{c}{\text{mM}} \cdot \frac{N_A}{10^{27}\,\text{nm}^3}$.

Mass conservation requires that:

$$(V - V_c)\rho_{R,d} + V_c\rho_{R,c} = V\rho_R$$

and

$$(V - V_c)\rho_{P,d} + V_c\rho_{P,c} = V\rho_P \tag{11}$$

leaving $V_c$ and $\rho_{R,c}$, and $\rho_{P,c}$ as independent variables to be determined for a given system in case of phase separation.

In general, the following scenarios are possible:

1. A fully disperse system, where there is no high-density condensate, that is, $V_c = 0$, $\rho_{R,d} = \rho_R$, $\rho_{P,d} = \rho_P$, $\rho_{R,c} = 0$, and $\rho_{P,c} = 0$;
2. A fully condensed system, that is, $\rho_{R,d} = 0$, $\rho_{P,d} = 0$, $\rho_{R,c} = \rho_R$, and $\rho_{P,c} = \rho_P$;
3. A phase-separated system with coexistence of dilute and condensed phases for both R and P particles, that is, $\rho_{R,d} > 0$ and $\rho_{P,d} > 0$;
4. A phase-separated system where only R particles coexist between dilute and condensed phases, that is, $\rho_{R,d} > 0$, $\rho_{P,d} = 0$, and $\rho_{P,c} = \rho_P$;
5. A phase-separated system where only P particles coexist between dilute and condensed phases, that is, $\rho_{R,d} = 0$, $\rho_{P,d} > 0$, and $\rho_{R,c} = \rho_R$.

Which of these possible scenarios is assumed, depends on the total free energy of the system.

In order to determine the total free energy of the system, we begin by estimating the chemical potential for a particle either in the dilute (d) and condensed (c) phase from enthalpies and entropies according to:

$$\mu_{R,d} = \Delta h_{R,d} - T\Delta s_{R,d}$$

$$\mu_{P,d} = \Delta h_{P,d} - T\Delta s_{P,d}$$

$$\mu_{R,c} = \Delta h_{R,c} - T\Delta s_{R,c}$$

$$\mu_{P,c} = \Delta h_{P,c} - T\Delta s_{P,c} \tag{12}$$

In the following, only terms for the dilute phase are given. The terms for the condensed phase are obtained in an equivalent manner.

The enthalpy terms are decomposed into interactions of R-R, P-P, and R-P pairs:

$$\Delta h_{R,d} = \Delta h_{d,RR} + \Delta h_{d,RP} \tag{13}$$

$$\Delta h_{P,d} = \Delta h_{d,PR} + \Delta h_{d,PP} \tag{14}$$

Each pairwise interaction energy is estimated from the coarse-grained interaction potential by assuming a spherically symmetric distribution of particles but modulated as a function of distance according to radial distribution function extracted from simulations for each pair. This amounts to convoluting the pairwise interaction potential $U$ (see *Equation 3*) with scaled volume- and density-normalized radial distribution functions $\hat{g}$ as follows:

$$\Delta h_{d,RR} = \frac{1}{2} \rho_{R,d} \int_V \hat{g}_{RR,d}(r) U_{RR}(r) d^3 r$$

$$= 2\pi \rho_{R,d} \int_0^{r_{max}} \hat{g}_{RR,d}(r) U_{RR}(r) r^2 dr \tag{15}$$

$$\Delta h_{d,PP} = 2\pi \rho_{P,d} \int_0^{r_{max}} \hat{g}_{PP,d}(r) U_{PP}(r) r^2 dr \tag{16}$$

$$\Delta h_{d,RP} = 2\pi \rho_{P,d} \int_0^{r_{max}} \hat{g}_{RP,d}(r) U_{RP}(r) r^2 dr \tag{17}$$

$$\Delta h_{d,PR} = 2\pi \rho_{R,d} \int_0^{r_{max}} \hat{g}_{PR,d}(r) U_{PR}(r) r^2 dr \tag{18}$$

where the factor 1/2 corrects for double-counted self-interactions.

Different radial distribution functions were used for dilute and condensed environments (*Figure 2—figure supplement 10*). The $g(r)$ functions extracted from the simulations were truncated at 20 nm and set to a constant value of 1 for larger radii to remove finite-size artifacts. Although the $g(r)$ functions were determined from simulations with specific sizes $r_{R,MD}$, $r_{P,MD}$ of the R and P particles, other particle sizes could be considered by scaling the radial dependence of the $g(r)$ functions according to the ratios $r_R/r_{R,MD}$, $r_P/r_{P,MD}$, and $(r_R+r_P)/(r_{R,MD}+r_{P,MD})$ for R-R, P-P, and R-P interactions. The upper integration limit $r_{max}$ was set to 100 nm for all interactions. At that radius and above, $U(r)$ is negligible for the range of radii and charges considered here. With the fixed integration limit, the integrals in *Equations 15 to 18* vary only with the charges and radii of particles R and P, and, thus, they are independent of particle concentrations. Then, the enthalpy contributions can be written as:

$$\Delta h_{d,RR} = \rho_{R,d} x_{d,RR} \tag{19}$$

$$\Delta h_{d,PP} = \rho_{P,d} x_{d,PP} \tag{20}$$

$$\Delta h_{d,RP} = \rho_{P,d} x_{d,RP} \tag{21}$$

$$\Delta h_{d,PR} = \rho_{R,d} x_{d,PR} \tag{22}$$

where the $x$ values represent the integrals in *Equations 15 to 18* multiplied by $2\pi$.

The entropy term was calculated based on the change of concentration in either dilute or condensed phases relative to the concentration in a fully disperse, non-separated system, which is the total system concentration, that is, for the dilute phase:

$$\Delta s_{R,\ d} = R\log\left(\frac{c_R}{c_{R,d}}\right) = R\log\left(\frac{\rho_R}{\rho_{R,d}}\right) \tag{23}$$

$$\Delta s_{P,\ d} = R\log\left(\frac{c_P}{c_{P,d}}\right) = R\log\left(\frac{\rho_P}{\rho_{P,d}}\right) \tag{24}$$

where R is the universal gas constant. In estimating the entropy for the condensed phase, the finite volumes of the R and P particles were subtracted from the condensed phase volume $V_c$:

$$\Delta s_{R,\ c} = R\log\left(\frac{\rho_R}{\rho_{R,c}} \cdot \left(1 - \left(\rho_{R,c}V_R + \rho_{P,c}V_P\right)\right)\right) \tag{25}$$

$$\Delta s_{P,\ c} = R\log\left(\frac{\rho_P}{\rho_{P,c}} \cdot \left(1 - \left(\rho_{R,c}V_R + \rho_{P,c}V_P\right)\right)\right) \tag{26}$$

with the molecular volumes calculated from the radii of the spherical R and P particles:

$$V_R = \frac{4\pi}{3}r_R^3$$

and

$$V_P = \frac{4\pi}{3}r_P^3 \tag{27}$$

Coexistence of the dilute and condensed phases assumes equilibrium, that is:

$$\mu_{R,d} = \mu_{R,c} \tag{28}$$

$$\mu_{P,d} = \mu_{P,c} \tag{29}$$

In scenario (3), both, *Equations 28 and 29*, have to be satisfied simultaneously. For scenario (4), only *Equation 28* needs to be satisfied under the condition that $\rho_{P,d} = 0$; and for scenario (5), only *Equation 29* has to be satisfied with $\rho_{R,d} = 0$.

Solutions in terms of $\rho_{R,d}$, $\rho_{P,d}$, $\rho_{R,c}$, $\rho_{P,c}$, and $V_c$ were determined numerically by scanning $V_c$ and solving for the densities in the dilute phase (the densities in the condensed phase follow from *Equation 11*).

*Equation 28* combined with *Equations 11, 12, 13, 19, 21, 23, and 25* gives the following:

$$0 = \mu_{R,d} - \mu_{R,c} \tag{30}$$

$$= \Delta h_{R,\ d} - T\Delta s_{R,d} - \Delta h_{R,\ c} + T\Delta s_{R,c}$$

$$= \rho_{R,d}x_{d,RR} + \rho_{P,d}x_{d,RP} - \rho_{R,c}x_{c,RR} - \rho_{P,c}x_{c,RP} + TR\log\left(\frac{\rho_{R,d}}{\rho_{R,c}} \cdot \left(1 - \left(\rho_{R,c}V_R + \rho_{P,c}V_P\right)\right)\right)$$

$$= \rho_{R,d}\left(x_{d,RR} + \frac{V - V_c}{V_c}x_{c,RR}\right) + \rho_{P,d}\left(x_{d,RP} + \frac{V - V_c}{V_c}x_{c,RP}\right) - \frac{V}{V_c}\left(\rho_R x_{c,RR} + \rho_P x_{c,RP}\right)$$

$$+ TR\log\left(\frac{V_c\rho_{R,d} - V_R\rho_{R,d}\left(V\rho_R - (V - V_c)\rho_{R,d}\right) - V_P\rho_{R,d}\left(V\rho_P - (V - V_c)\rho_{P,d}\right)}{V\rho_R - (V - V_c)\rho_{R,d}}\right)$$

$$= f_R\left(\rho_{R,d}, \rho_{P,d}, V_c\right) \tag{31}$$

An analogous function $f_P(\rho_{R,d}, \rho_{P,d}, V_c)$ is obtained from *Equation 29*. There is no analytical solution, but $f_R(\rho_{R,d}, \rho_{P,d}, V_c) = 0$ and $f_P(\rho_{R,d}, \rho_{P,d}, V_c) = 0$ can be solved via the Newton-Raphson method given $V_c$ and either $\rho_{P,d}$ or $\rho_{R,d}$.

For scenario (4), $f_R(\rho_{R,d}, \rho_{P,d}, V_c) = 0$ was solved for different values of $V_c$ and $\rho_{P,d} = 0$; for scenario (5), $f_P(\rho_{R,d}, \rho_{P,d}, V_c) = 0$ was solved for values of $V_c$ and $\rho_{R,d} = 0$. For scenario (3), $\rho_{R,d}$ was scanned as well and the value of $\rho_{P,d}$ was determined for given values of $V_c$ and $\rho_{R,d}$ by first solving $f_R(\rho_{R,d}, \rho_{P,d}, V_c) = 0$. The resulting value of $\rho_{P,d}$ was then used with $V_c$ to solve $f_P(\rho_{R,d}, \rho_{P,d}, V_c) = 0$ for a refined value of $\rho_{R,d}$.

Mathematically possible solutions include cases where the volume fractions in the cluster exceed what is physically realistic inside the condensed state. In order to exclude such solutions, it was required that the combined macromolecular volume in the condensed phase is less than 30% of the total volume of the condensed phase, that is,:

$$\rho_{R,c} V_R + \rho_{P,c} V_P < 0.3 \tag{32}$$

We note that most final solutions were found at the 30% vol fraction limit, since the theory did not directly account for volume exclusion between individual molecules and found a gain in energy at higher particle densities. However, similar results were obtained with maximal macromolecular volume fractions according to *Equation 32* in a range of 20–40%. The value of 30% was ultimately arrived at by optimal agreement between theory and experiment for the concentration-dependent phase separation between RNA and trypsin shown in *Figure 6*.

The total system energy is calculated according to:

$$\Delta G = \mu_{R,d} \cdot (V - V_c)\rho_{R,d} + \mu_{R,c} \cdot V_c \rho_{R,c} + \mu_{P,d} \cdot (V - V_c)\rho_{P,d} + \mu_{P,c} \cdot V_c \rho_{P,c} - TS_{mix} \tag{33}$$

where $S_{mix}$ is the overall mixing entropy according to the ratio of particles R and P in the dilute and condensed phases according to:

$$S_{mix} = S_{mix,d} + S_{mix,c} \tag{34}$$

$$S_{mix,d} = R(V - V_c)\left(\rho_{R,d}\log\frac{\rho_{R,d}}{\rho_{R,d} + \rho_{P,d}} + \rho_{P,d}\log\frac{\rho_{P,d}}{\rho_{R,d} + \rho_{P,d}}\right) \tag{35}$$

$$S_{mix,c} = RV_c\left(\rho_{R,c}\log\frac{\rho_{R,c}}{\rho_{R,c} + \rho_{P,c}} + \rho_{P,c}\log\frac{\rho_{P,c}}{\rho_{R,c} + \rho_{P,c}}\right) \tag{36}$$

For the five scenarios described above, total free energies were then calculated as follows:
(1) Disperse:

$$\Delta G_1 = \mu_{R,disperse} \cdot V\rho_R + \mu_{P,disperse} \cdot V\rho_P - TRV\left(\rho_R\log\frac{\rho_R}{\rho_R + \rho_P} + \rho_P\log\frac{\rho_P}{\rho_R + \rho_P}\right) \tag{37}$$

where $\mu_{\frac{R}{P},disperse}$ were calculated according to *Equations 12 to 18* using RDFs from the disperse phase extracted from our molecular dynamics simulations before condensates started to form.
(2) Condensed:

$$\Delta G_2 = \mu_{R,c} \cdot V\rho_R + \mu_{P,c} \cdot V\rho_P - TRV_c\left(\rho_R\log\frac{\rho_R}{\rho_R + \rho_P} + \rho_P\log\frac{\rho_P}{\rho_R + \rho_P}\right) \tag{38}$$

(3) R and P in phase coexistence:

$$\Delta G_3 = \mu_{R,c} \cdot V\rho_R + \mu_{P,c} \cdot V\rho_P \tag{39}$$

$$-TR(V - V_c)\left(\rho_{R,d}\log\frac{\rho_{R,d}}{\rho_{R,d} + \rho_{P,d}} + \rho_{P,d}\log\frac{\rho_{P,d}}{\rho_{R,d} + \rho_{P,d}}\right)$$

$$-TRV_c\left(\rho_{R,c}\log\frac{\rho_{R,c}}{\rho_{R,c}+\rho_{P,c}}+\rho_{P,c}\log\frac{\rho_{P,c}}{\rho_{R,c}+\rho_{P,c}}\right)$$

since $\mu_{R,c}=\mu_{R,d}$ and $\mu_{P,c}=\mu_{P,d}$.
(4) R in phase coexistence, $\rho_{P,d}=0$:

$$\Delta G_4=\mu_{R,c}\cdot V\rho_R+\mu_{P,c}\cdot V\rho_P-TRV_c\left(\rho_{R,c}\log\frac{\rho_{R,c}}{\rho_{R,c}+\rho_{P,c}}+\rho_{P,c}\log\frac{\rho_{P,c}}{\rho_{R,c}+\rho_{P,c}}\right) \tag{40}$$

(5) P in phase coexistence, $\rho_{R,d}=0$:

$$\Delta G_5=\mu_{R,c}\cdot V\rho_R+\mu_{P,c}\cdot V\rho_P-TRV_c\left(\rho_{R,c}\log\frac{\rho_{R,c}}{\rho_{R,c}+\rho_{P,c}}+\rho_{P,c}\log\frac{\rho_{P,c}}{\rho_{R,c}+\rho_{P,c}}\right) \tag{41}$$

The scenario with the overall lowest free energy was then considered to be the predicted state.

A program implementing this model is available at http://github.com/feiglab/phasesep; copy archived at swh:1:rev:24890516a822b917b76a2730ced19839acbaec3d

## Experimental materials and methods

The J345 RNA sequence was synthesized and deprotected by Dharmacon (Horizon Discovery Group), both with and without Cy3 or Cy5 on the 3' end. The 47-base sequence is GCAGCAGG-GAACUCACGCUUGCGUAGAGGCUAAGUGCUUCGGCACAGCACAAGCCCGCUGCG.

All measurements were made using the buffer used by Bonneau and Legault for structure determination of this sequence, 10 mM sodium cacodylate (pH 6.5), 50 mM NaCl,. 05% sodium azide, 5 mM MgCl$_2$. Equine liver trypsin, equine alcohol dehydrogenase, bovine lactic dehydrogenase, equine myoglobin, hen egg lysozyme, and bovine serum albumin were obtained from Sigma-Aldrich and used without further modification.

### Microscopy

Confocal microscopy images were obtained on a Nikon A1 scanning confocal microscope with 100x magnification. The excitation wavelength was 561 nm and detection was set for Cy3 fluorescence using a GaAsP detector. The diffraction-limited spatial resolution is 260 nm. Images were processed with ImageJ and modified only for contrast and brightness. Images were cropped and enlarged to aid observation of the smallest features.

### Dynamic light scattering

The size distribution of the protein-RNA complexes were measured using a dynamic light scattering (DLS) machine (Zetasizer nano series from Malvern company) at room temperature. The samples were mixed freshly before each experiment and all measurements were repeated three times in a single run and the corresponding average results were reported. A Helium Neon laser with a wavelength of 632 nm was used for the size distribution analysis.

The central observable of dynamic light scattering (DLS) experiments consists of time-dependent scattering intensity correlation functions $g_2(\tau)$ that are related to electric field correlation functions $g_1(\tau)$ according to:

$$g_2(\tau)-1=g_1(\tau)^2 \tag{42}$$

In case of a monodisperse solution with particles of a diameter $d$, a single exponential decay is observed with:

$$g_1(\tau;d)=e^{-2q^2D(d)\tau} \tag{43}$$

with the wave vector

$$q=\frac{4\pi n}{\lambda}\sin\left(\frac{\theta}{2}\right) \tag{44}$$

and the diffusion according to Stokes-Einstein:

$$D(d) = \frac{k_B T}{6\pi\eta d} \tag{45}$$

where $n$ is the refractive index of the solvent medium (i.e. 1.335), $\lambda$ is the wavelength of the incident laser light (i.e. 633 nm), $\theta$ is the scattering angle (i.e. 173°), $k_B$ is the Boltzmann constant, T is the temperature (i.e. 298 K), and $\eta$ is the viscosity of the solvent (i.e. 0.8882 cP).

The samples we considered were clearly polydisperse, requiring the fit of multiple exponential decays. Moreover, from previous studies and simulations, we expect that at the smallest particle sizes there is an exponential decay of particle sizes due to dynamic cluster formation in the dilute phase (*Nawrocki et al., 2019b*, *von Bülow et al., 2019*). Therefore, we fit the experimental data (i.e. $g_2(\tau) - 1$) to the following function:

$$g_2(\tau) - 1 = g_1(\tau)^2 \approx \sum_{i=1}^{10} a_c^2 e^{-\frac{2i}{t_c}} g_1^2(\tau; d_c) + \sum_{i=1}^{4} a_i^2 g_1^2(\tau; d_i) \tag{46}$$

Consequently, the parameters of the numerical fits were the size of the smallest particle, $d_c$, its contribution, $a_c$, decreasing according to the decay 'time' $t_c$, and an additional up to four discrete sizes $d_i$ with contributions $a_i$.

Using gnuplot, version 5.2, we fit the function according to *Equation 46* to individual correlation functions as well to an average that was obtained after normalizing individual functions.

## FRET spectroscopy

Fluorescence spectra were obtained with PTI Q4 fluorimeter, excited at 475 nm and emission observed between 525 and 700 nm. The concentration of Cy3-labeled RNA and Cy5-labeled RNA were kept constant at 8 µM and 42 µM, respectively, with the unlabeled concentration varied from 0 to 0.5 mM. The low concentration of labeled RNA limits the possibility of self-quenching but also limits the detection of very small clusters.

The normalized FRET ratio was calculated from the total intensity between 525 and 650 nm for the donor and 650 and 700 nm for the acceptor,

$$FRET = \frac{I_A}{I_D + I_A}. \tag{47}$$

In the absence of protein, the RNA exhibits some baseline transfer, likely due to transient interactions between the dyes, leading to a background FRET level of ~0.24. Upon the addition of protein above the threshold concentration, the mixture is visibly turbid.

FRET efficiencies for mixtures of RNA and proteins at different concentrations were estimated from the predicted amount of RNA inside and outside the condensates as follows:

The theory described above predicts phase separation with the densities of RNA in the dilute and condensed phases given as $\rho_{R,d}$ and $\rho_{R,c}$. From the densities the concentration of RNA in the dilute ([$R_d$]) and condensed ([$R_c$]) phases with respect to the total volume is obtained as follows:

$$[R_d] = \rho_{R,d} \cdot \frac{V - V_c}{V} \tag{48}$$

$$[R_c] = \rho_{R,c} \cdot \frac{V_c}{V} \tag{49}$$

A fraction of RNA is labeled with fluorophores. The total concentration of labeled RNA is denoted as [F]; the concentration in the dilute and condensed phases, again with respect to the total system volume, is denoted as [$F_d$] and [$F_c$], respectively. Then:

$$[F] = [F_c] + [F_d] \tag{50}$$

and

$$[R_d] = [U_d] + [F_d] \tag{51}$$

$$[R_c] = [U_c] + [F_c] \tag{52}$$

where $[U_d]$ and $[U_c]$ are the concentrations of unlabeled RNA in the dilute and condensed phases.

We further make an assumption that there is an equilibrium of labeled RNA to exchange between the dilute and condensed phases while maintaining the overall ratio of RNA between the two phases:

$$[F_d] + [U_c] \leftrightarrow [F_c] + [U_d] \tag{53}$$

with the equilibrium constant $K$ given as:

$$K = \frac{[F_c][U_d]}{[F_d][U_c]} \tag{54}$$

Because of the hydrophobic character of the FRET labels we expect that labeled RNA has an affinity for the less-hydrated condensate, that is, $K > 1$.

*Equations 50, 51, 52, 54* can be solved for $[F_c]$ as a function of $[R_d]$, $[R_c]$, $[F]$, and $K$ to give the fraction of labeled RNA in the condensate as:

$$f = \frac{[F_c]}{[F]} \tag{55}$$

Based on the resulting value of $f$, FRET efficiencies $E$ were then estimated according to:

$$E = E_0(1 - f) + E_c f \tag{56}$$

where $E_0$ and $E_c$ are the FRET efficiencies at zero protein concentration and in the condensed phase, respectively. $E_0$ was taken from experiment and $E_c$ was estimated by convoluting the distribution of minimum RNA-RNA distances in the condensed phase extracted from the simulations with $1/(1+(r/r_0)^6)$, where $r$ is the distance between RNA molecules and $r_0$ is a constant that depends on the fluorescence label and additional factors such as the anisotropy of the orientational sampling and the index of diffraction of the medium.

We applied this formalism to interpret the FRET experiments on trypsin based on predicted RNA fractions in the condensed phase (*Figure 9—figure supplement 7*) using the minimum distance distribution of RNA shown in *Figure 2—figure supplement 11*. We took $E_0 = 0.24$ from experiment and found good agreement between experiment and theory for $r_0 = 4.10$ nm and $K = 100$ (*Figure 7*). We note that the value $r_0 = 4.10$ nm is lower than typical values assumed for the Cy3-Cy5 pair (*Murphy et al., 2004*), but the condensed state differs from typical solution conditions, whereas the spherical models used here allow only very approximate estimates of the true donor-acceptor distances and neglect orientational dependence in fluorescent energy transfer (*Iqbal et al., 2008*).

## NMR spectroscopy

NMR spectra were acquired at a $^1$H frequency of 600 MHz on a Varian 600 MHz spectrometer with a room-temperature probe. Solvent was suppressed with a gradient 1–1 echo sequence. Samples were prepared in 90% $H_2O$, 10% $D_2O$ in the buffer described above with DSS as an internal chemical shift reference. 16 k points were acquired with a 1 s recycle delay and a total acquisition time of approximately 1 hr per spectrum. RNA concentrations were 300 μM for J345 only and 135–140 μM for RNA-protein samples; protein concentrations were around 150 μM; the RNA-only spectrum was scaled to account for the differing concentration. Spectra were processed with zero-filling to 32 k and a 5 Hz exponential window function.

## Circular dichroism spectroscopy

Circular dichroism measurements were made using an Applied Photophysics Chirascan spectrometer. All measurements were made using a 0.1 mm pathlength cuvette at room temperature.

## Acknowledgements

This study was funded by the National Science Foundation (MCB 1817307, to MF and LJL; MCB 2018296, to CGH) and the National Institutes of Health (R35 GM126948, to MF). Computational resources at the Institute for Cyber-Enabled Research/High Performance Computing Cluster (ICER/ HPCC) at Michigan State University were used.

## Additional information

### Funding

| Funder | Grant reference number | Author |
|---|---|---|
| National Institutes of Health | R35 GM126948 | Bercem Dutagaci Grzegorz Nawrocki Michael Feig |
| National Science Foundation | MCB 1817307 | Lisa J Lapidus Michael Feig |
| National Science Foundation | MCB 2018296 | Charles G Hoogstraten |

The funders had no role in study design, data collection and interpretation, or the decision to submit the work for publication.

### Author contributions

Bercem Dutagaci, Conceptualization, Data curation, Software, Formal analysis, Validation, Investigation, Visualization, Methodology, Writing - original draft, Writing - review and editing; Grzegorz Nawrocki, Data curation, Software, Investigation, Methodology, Writing - review and editing; Joyce Goodluck, Investigation, Writing - review and editing; Ali Akbar Ashkarran, Resources, Data curation, Investigation, Writing - review and editing; Charles G Hoogstraten, Conceptualization, Resources, Data curation, Formal analysis, Investigation, Writing - review and editing; Lisa J Lapidus, Conceptualization, Resources, Data curation, Formal analysis, Supervision, Funding acquisition, Validation, Investigation, Visualization, Methodology, Writing - original draft, Project administration, Writing - review and editing; Michael Feig, Conceptualization, Resources, Data curation, Software, Formal analysis, Supervision, Funding acquisition, Validation, Investigation, Visualization, Methodology, Writing - original draft, Project administration, Writing - review and editing

### Author ORCIDs

Bercem Dutagaci (iD) http://orcid.org/0000-0003-0333-5757
Michael Feig (iD) https://orcid.org/0000-0001-9380-6422

### Decision letter and Author response

Decision letter https://doi.org/10.7554/eLife.64004.sa1
Author response https://doi.org/10.7554/eLife.64004.sa2

## Additional files

### Supplementary files

• Supplementary file 1. Components in cytoplasmic model, CG parameters, and self-diffusion. Macromolecular components in a previously established model of the bacterial cytoplasm of *Mycoplasma genitalium* (*Nguemaha and Zhou, 2018*) with their molecular properties and corresponding CG model parameters. Self-diffusion rates extracted from the CG simulations are reported for different parts of the trajectory and different parts of the system (tRNA clusters, RP clusters, outside clusters).

• Supplementary file 2. Five-component model simulations. List of the five-component model simulations with molecular compositions, CG parameters, and simulation conditions.

• Transparent reporting form

## Data availability

All experimental data generated and analyzed during this study are included in the manuscript and supporting files.

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
