## [Decision Letter]

**Acceptance summary:**

The driving forces behind the formation of membrane-less organelles within cells have been of significant interest. The results of this article suggest that non-specific electrostatic interactions primarily control phase separation of RNA/protein condensation in a cytoplasmic-like environment. The results suggest that phase separation occurs readily in simulations, and the simulations reproduce experimental results of binary mixtures of proteins and RNA. The predictions made by a theory developed by the authors to explain the experimental results are in good agreement with the simulation data. The combination of the different techniques is commendable, and it is anticipated that the results presented will stimulate additional studies.

**Decision letter after peer review:**

Thank you for submitting your article "Charge-Driven Phase Condensation of RNA and Proteins Suggests Broad Role of Phase Separation in Cytoplasmic Environments" for consideration by *eLife*. Your article has been reviewed by three peer reviewers, including Donald Hamelberg as the Reviewing Editor and Reviewer #1, and the evaluation has been overseen by José Faraldo-Gómez as the Senior Editor

The reviewers have discussed the reviews with one another and the Reviewing Editor has drafted this decision to help you prepare a revised submission.

Summary:

The manuscript by Dutagaci et al., describes a combination of coarse-grained (CG) simulations, theory, and experiment to examine the possibility for phase separation to occur in mixed protein-RNA systems that are intended to mimic cytoplasmic conditions. The authors develop a theory to predict and explain experimental results of binary systems. The authors find that phase separation occurs readily in simulations in which cytoplasmic components are modeled as spheres, and they then show that the same simulation models do a reasonable job of reproducing experimental microscopy results obtained for simple binary mixtures of proteins and RNA. Especially encouraging is that good qualitative correspondence is obtained between simulation and experiment with regard to which proteins form condensates with a model small RNA (J345); the predictions made by the theory are also shown to be in quite good agreement with the simulation data. However, it is not completely clear to the reviewers how much the condensates maintain a liquid character. The paper is well written, the results are interesting, and the attempt to combine different techniques is commendable.

The work will be of interest to many in the field. The results would stimulate additional studies and experiments by the authors and others. Below are recommended revisions to improve the manuscript and strengthen the conclusions of the work.

Essential revisions:

1) It is very clear from the beautiful simulation snapshots shown in Figure 1 that phase separation occurs in the CG simulations of the authors' cytoplasm model. There is no question that this is a very interesting and arresting result. What is less convincing, however, is how strong the evidence is in support of the statement that the separated phases seen in the simulations are "liquid condensates". The nice movie provided by the authors gives an impression of two gel phases rather than two liquid phases; gel-like behavior would also seem to be a likely outcome given the absence of hydrodynamic interactions in the simulations. In support of liquid-like behavior, the authors mention that the macromolecular concentrations are consistent with liquid estimates and that, for the tRNA-dominated condensate at least, the RDF has few clear peaks (Figure 1—figure supplement 3); the latter feature, however, could be caused by small displacements of many tRNAs about nearly-fixed positions within the condensates. The authors also argue that their measured translational diffusion coefficients (Dtr) indicate liquid-like behavior. But these are obtained from fits to MSD data with lag times of only up to 20 ns, and it is clear from the plots that the lines are deviating from linearity as the lag time increases up to 50 ns. If the authors were to instead measure Dtr with a series of much longer lag times (e.g. up to 1 us: very feasible given the 1 ms simulation time and the number of molecules in each condensate) then I think that they might find that their Dtr estimates would drop lower and lower as the lag time increases. If so, wouldn't that indicate that the condensates are gels rather than liquids? The phase separation observed by the authors is a sufficiently interesting result to report, without needing to go one step further and say that the observed phases are liquid-like. If the authors really want to argue that the phases they observe are liquid then they need to provide stronger evidence; if not, wouldn't it would be better to just say "phase separation" throughout the manuscript and leave it at that?

2) Related to the point above, the experiments undoubtedly confirm the formation of condensates between RNA and positively charge proteins. However, the nature of the condensates is ambiguous for majority of the RNA/protein systems studied. Surprisingly, it appears that only one of these condensates possibly maintained a liquid character, contrary to simulation results of the multicomponent systems. In general, are two component systems capable of forming liquid phase condensates? What is known?

3) The simulated cytoplasm model is a reduced representation of an atomistic model previously reported by the authors in this journal. The trade-off made here is that the reduced representation allows much longer timescales to be simulated and this may well be the key to the interesting behavior uncovered by the authors. The model itself, however, is very reminiscent of cytoplasm models previously reported in the literature by other groups: starting with seminal work by Bicout and Field, (1996), but continuing with works by Ridgway et al., (2008), McGuffee and Elcock, (2010), Ando and Skolnick, (2010), Qang and Cheung, (2012), Xu et al., (2013), Hasnain etr al., (2014), Trovato and Tozzini, (2014) and probably others. Of these studies, only the Ando and Skolnick paper is cited here. Some of those earlier studies use models *very* similar to that used here, while others employ models that are much more structurally detailed (though not having the explicit solvent used in the authors' previous atomistic MD simulations). While the present study is exciting in both its timescale and its findings, omitting references to those other papers does readers a disservice and makes the authors look ungenerous to others working in the field. A paragraph should be added to the Introduction and Discussion section that cites and acknowledges the good simulation work done by others that came before this study.

4) Given the potential implications for "real life" in vivo, the authors should also provide some deeper discussion of the potential shortcomings of their CG cytoplasm model, e.g. with regard to: (a) components that are only very roughly treated (e.g. the "mRNA" is 6 copies of a 100-nucleotide RNA, which isn't enough to code for anything interesting), (b) components that might be important but that have been left out (DNA? metabolites?), and/or (c) components for which a sphere might not be a particularly good model (see mRNA above).

5) The authors account for bound counterions by modifying the effective charge on each sphere using Equation 5 or Equation 6. It is unclear where these equations came from. Were they made up by the authors (e.g. Equation 6?) or where they obtained from another source?

6) The algorithm used in the simulations should be clarified. As written, it sounds like conventional molecular dynamics (MD) was used, which would be an odd choice since it would mean that the macromolecules would move ballistically between collisions. But in the Materials and methods section we are told that a Langevin thermostat was used in the simulations, which would make them stochastic dynamics (SD) simulations instead.

7) Additional text would be helpful for the legend to Figure 3 and/or Materials and methods section outlining how each of the data points on the phase diagrams in panels A-D were obtained: after multiple readings of this section of the paper it is reasonably clear about where the fit-lines come from, but it is unclear on the much more basic issue of how the actual data points were obtained.

8) In fitting their theory to their simulation data, the authors note that "Mathematically possible solutions include cases where the volume fractions in the cluster exceed what is physically realistic". They therefore restrict their solutions to ones in which the total macromolecular volume is less than 30%. This doesn't seem unreasonable, but the 30% number is somewhat arbitrary. The authors should say how many of their final solutions ended up with a volume of 29.999% as this would tell whether the theory really wanted to settle on a quite different answer than the one ultimately accepted by the authors.

9) It may be worth elaborating on the "… highly ordered arrangement in the RP condensates." It was recently shown that a balance of homotypic and heterotypic interactions might lead to structured condensates (10.1093/nar/gkaa1099).

---

## [Author Response]

Essential revisions:1) It is very clear from the beautiful simulation snapshots shown in Figure 1 that phase separation occurs in the CG simulations of the authors' cytoplasm model. There is no question that this is a very interesting and arresting result. What is less convincing, however, is how strong the evidence is in support of the statement that the separated phases seen in the simulations are "liquid condensates". The nice movie provided by the authors gives an impression of two gel phases rather than two liquid phases; gel-like behavior would also seem to be a likely outcome given the absence of hydrodynamic interactions in the simulations. In support of liquid-like behavior, the authors mention that the macromolecular concentrations are consistent with liquid estimates and that, for the tRNA-dominated condensate at least, the RDF has few clear peaks (Figure 1—figure supplement 3); the latter feature, however, could be caused by small displacements of many tRNAs about nearly-fixed positions within the condensates. The authors also argue that their measured translational diffusion coefficients (Dtr) indicate liquid-like behavior. But these are obtained from fits to MSD data with lag times of only up to 20 ns, and it is clear from the plots that the lines are deviating from linearity as the lag time increases up to 50 ns. If the authors were to instead measure Dtr with a series of much longer lag times (e.g. up to 1 us: very feasible given the 1 ms simulation time and the number of molecules in each condensate) then I think that they might find that their Dtr estimates would drop lower and lower as the lag time increases. If so, wouldn't that indicate that the condensates are gels rather than liquids? The phase separation observed by the authors is a sufficiently interesting result to report, without needing to go one step further and say that the observed phases are liquid-like. If the authors really want to argue that the phases they observe are liquid then they need to provide stronger evidence; if not, wouldn't it would be better to just say "phase separation" throughout the manuscript and leave it at that?

Thank you for the interesting comments. The question about the internal dynamics of the condensates is indeed important. We believe that we found evidence that the internal dynamics of molecules in the condensates is retained at least to some extent, but we agree that the exact delineation between liquid, gel, and other types of condensed phases is challenging and that it may be better to focus largely on ‘phase separation’ as the main observation described here.

With respect to the specific suggestion of considering longer lag phases in the diffusion analysis from the simulations, we followed that idea and analyzed MSD curves with lag times up to 20 µs. As the reviewer suspected, MSD vs. lag time does not follow a linear trend. Instead, MSD values level off at values close to 400 nm^2^ at around 10 µs as shown in Author response image 1:

We believe that this observation is indicative of spherical confinement within the condensates rather than indicating gel-like behavior. Based on previous theoretical work (cf. e.g. Bickel, T. A note on confined diffusion*. Physica A* 2007, 377:24–32) an MSD value of 400 nm^2^ at the long-time limit is consistent with Brownian diffusion confined by a sphere with a radius of about 18 nm, which is quite close to the radius of the condensates. We believe that this indicates that at least the tRNA-containing condensates are indeed liquid-like in the CG simulations. For a gel phase, we would have expected much smaller limiting MSD values on shorter time scales and evidence of more complex diffusive behavior on longer time scales.We acknowledge that the CG simulations suffer from a lack of properly accounting for hydrodynamic interactions whereas diffusion within spherical confinement has been studied before. Moreover, experimental validation of liquid-like behavior is limited to the RNA-trypsin system. Therefore, we followed the suggestion to deemphasize discussing the formation of “liquid” condensates. We retained results from simulations and experiments that we believe provide a valuable characterization of biomolecular dynamics inside the condensates based on short-time diffusion, but we did not elaborate further on the longer-time behavior dominated by confinement.

2) Related to the point above, the experiments undoubtedly confirm the formation of condensates between RNA and positively charge proteins. However, the nature of the condensates is ambiguous for majority of the RNA/protein systems studied. Surprisingly, it appears that only one of these condensates possibly maintained a liquid character, contrary to simulation results of the multicomponent systems. In general, are two component systems capable of forming liquid phase condensates? What is known?

We agree that the exact nature of the condensates is not completely clear, primarily due to the small sizes observed for most systems. We did find what appears to be clearly liquid character for the RNA-trypsin system, but it is unclear how general that observation is for other RNA-protein condensates and what factors may determine the internal dynamics of other condensates. We also note that much of the “LLPS” literature is not very precise in exactly defining what constitutes “liquid” behavior. While we do not have more insight beyond the data we are presenting here, this is clearly a topic warranting further studies, for the systems described here as well as biologically relevant condensates described in other contexts.

3) The simulated cytoplasm model is a reduced representation of an atomistic model previously reported by the authors in this journal. The trade-off made here is that the reduced representation allows much longer timescales to be simulated and this may well be the key to the interesting behavior uncovered by the authors. The model itself, however, is very reminiscent of cytoplasm models previously reported in the literature by other groups: starting with seminal work by Bicout and Field, (1996), but continuing with works by Ridgway et al., (2008), McGuffee and Elcock, (2010), Ando and Skolnick, (2010), Qang and Cheung, (2012), Xu et al., (2013), Hasnain et al., (2014), Trovato and Tozzini, (2014) and probably others. Of these studies, only the Ando and Skolnick paper is cited here. Some of those earlier studies use models *very* similar to that used here, while others employ models that are much more structurally detailed (though not having the explicit solvent used in the authors' previous atomistic MD simulations). While the present study is exciting in both its timescale and its findings, omitting references to those other papers does readers a disservice and makes the authors look ungenerous to others working in the field. A paragraph should be added to the Introduction and Discussion section that cites and acknowledges the good simulation work done by others that came before this study.

The reviewer is correct that our work follows similar previous works – with the important differences that the parametrization of our model is based on atomistic simulations apart from the longer time scales and larger spatial scales that are covered here. We agree that more extensive references of previous works are appropriate, and we added additional references as suggested. Otherwise, we are referring to two recent reviews that cover the modeling of cytoplasmic environments more broadly.

4) Given the potential implications for "real life" in vivo, the authors should also provide some deeper discussion of the potential shortcomings of their CG cytoplasm model, e.g. with regard to: (a) components that are only very roughly treated (e.g. the "mRNA" is 6 copies of a 100-nucleotide RNA, which isn't enough to code for anything interesting), (b) components that might be important but that have been left out (DNA? metabolites?), and/or (c) components for which a sphere might not be a particularly good model (see mRNA above).

The reviewer is correct that the CG cytoplasm model presented here is lacking many details of “real-life” environments. While we plan to add further aspect in future studies, some factors such as molecular details of different macromolecules remain difficult to treat with a computational model that allows access to the temporal and spatial scales necessary to describe phase separation processes in the cytoplasmic context.

We added a paragraph in the Discussion section elaborating on the shortcomings of our approach and outlining at least some avenues for how future cytoplasmic models at the CG level could be improved.

5) The authors account for bound counterions by modifying the effective charge on each sphere using Equation 5 or Equation 6. It is unclear where these equations came from. Were they made up by the authors (e.g. Equation 6?) or where they obtained from another source?

We are proposing these equations here to interpolate between previously proposed effective charges for moderately and highly charged biological macromolecules such as RNA and ribosomes due to counterion condensation. The two equations cover the range of estimates of effective charges found in past studies. The second expression (Equation 6) was ultimately found to lead to better agreement with the experimental data presented here and therefore we present most theoretical results based on that expression. A more extensive discussion is found in the Materials and methods section.

6) The algorithm used in the simulations should be clarified. As written, it sounds like conventional molecular dynamics (MD) was used, which would be an odd choice since it would mean that the macromolecules would move ballistically between collisions. But in the Materials and methods section we are told that a Langevin thermostat was used in the simulations, which would make them stochastic dynamics (SD) simulations instead.

We believe that “MD with a Langevin thermostat” and “Stochastic dynamics” describe the same methodology although different communities may prefer one or the other term. To avoid confusion, we added a sentence in the Materials and methods section to explicitly state that the simulations described here reflect stochastic dynamics.

7) Additional text would be helpful for the legend to Figure 3 and/or Materials and methods section outlining how each of the data points on the phase diagrams in panels A-D were obtained: after multiple readings of this section of the paper it is reasonably clear about where the fit-lines come from, but it is unclear on the much more basic issue of how the actual data points were obtained.

The data points are volume fractions of RNA inside and outside the condensates based on the number of RNA in the largest cluster (corresponding to the condensed state) and a volume estimated based on the overlap of atomic van der Waals volumes of the particles inside the largest cluster.

Additional information was provided in the text and in the legend to Figure 3 to explain this more clearly.

8) In fitting their theory to their simulation data, the authors note that "Mathematically possible solutions include cases where the volume fractions in the cluster exceed what is physically realistic". They therefore restrict their solutions to ones in which the total macromolecular volume is less than 30%. This doesn't seem unreasonable, but the 30% number is somewhat arbitrary. The authors should say how many of their final solutions ended up with a volume of 29.999% as this would tell whether the theory really wanted to settle on a quite different answer than the one ultimately accepted by the authors.

Most solutions did in fact end up with a volume fraction of 30% since the theory does not directly account for volume exclusion between individual molecules and mathematically optimal solutions typically resulted in physically unrealistic packing fractions. The theory results in terms of finding condensates or not do not change much when that threshold is varied within a reasonable range (i.e. 20-40%), but we note that the specific value of 30% was ultimately chosen to best describe the experimental data. We added text to explain this in more detail

9) It may be worth elaborating on the "… highly ordered arrangement in the RP condensates." It was recently shown that a balance of homotypic and heterotypic interactions might lead to structured condensates (10.1093/nar/gkaa1099).

Thank you for bringing this to our attention. We added a sentence to the Introduction and added a reference to this work.